# Disentangling and mitigating the impact of task similarity for continual learning

**Naoki Hiratani**
Department of Neuroscience
Washington University in St Louis
St Louis, MO 63110
`hiratani@wustl.edu`

## Abstract

Continual learning of partially similar tasks poses a challenge for artificial neural networks, as task similarity presents both an opportunity for knowledge transfer and a risk of interference and catastrophic forgetting. However, it remains unclear how task similarity in input features and readout patterns influences knowledge transfer and forgetting, as well as how they interact with common algorithms for continual learning. Here, we develop a linear teacher-student model with latent structure and show analytically that high input feature similarity coupled with low readout similarity is catastrophic for both knowledge transfer and retention. Conversely, the opposite scenario is relatively benign. Our analysis further reveals that task-dependent activity gating improves knowledge retention at the expense of transfer, while task-dependent plasticity gating does not affect either retention or transfer performance at the over-parameterized limit. In contrast, weight regularization based on the Fisher information metric significantly improves retention, regardless of task similarity, without compromising transfer performance. Nevertheless, its diagonal approximation and regularization in the Euclidean space are much less robust against task similarity. We demonstrate consistent results in a permuted MNIST task with latent variables. Overall, this work provides insights into when continual learning is difficult and how to mitigate it.

## 1 Introduction

Artificial neural networks surpass human capabilities in various domains, yet struggle with continual learning. These networks tend to forget previously learned tasks when trained sequentially—a problem known as catastrophic forgetting [44, 16, 23, 32]. This phenomenon affects not only supervised training of feedforward networks but also extends to recurrent neural networks [10], reinforcement learning tasks [30], and fine-tuning of large language models [41]. Many algorithms for mitigating catastrophic forgetting have been developed previously, including rehearsal techniques [48, 49, 59], weight regularization [30, 36, 67], and activity-gating methods [15, 54, 42, 56], among others [57, 50, 63, 21]. However, these methods often hinder forward and backward knowledge transfer [28, 39, 29], and thus it remains unclear how to achieve knowledge transfer and retention simultaneously.

A key factor for continual learning is task similarity. If two subsequent tasks are similar, there is a potential for a knowledge transfer from one task to another, but the risk of interference also becomes high [47, 28, 35, 11, 39]. The impact of task similarity on transfer and retention performance is particularly complicated because two tasks can be similar in different manners [65, 32]. Sometimes familiar input features need to be associated with novel output patterns, but at other times, novel input features need to be associated with familiar output patterns. Previous works observed that these

38th Conference on Neural Information Processing Systems (NeurIPS 2024).

two scenarios influence continual learning differently [35], yet the impact of the input and output similarity on knowledge transfer and retention has not been well understood. Moreover, it remains unknown how the task similarity interacts with algorithms for continual learning such as activity gating or weight regularization.

To gain insight into these questions, in this work, we investigate how transfer and retention performance depend on task similarity, task-dependent gating, and weight regularization in analytically tractable teacher-student models. Teacher-student models are simple, typically linear, neural networks in which the generative model of data is specified explicitly by the teacher network [18, 66, 3]. These models have provided tremendous insights into generalization property [55, 45, 19, 1], convergence rate [61, 58, 38], and learning dynamics [51, 52, 4, 25] of neural networks, due to their analytical tractability. Several works also studied continual learning using teacher-student settings [2, 35, 27, 11, 20, 37, 13] (see Related works section for details).

We develop a linear teacher-student model with a low-dimensional latent structure and analyze how the similarity of input features and readout patterns between tasks affect continual learning. We show analytically that a combination of low feature similarity and high readout similarity is relatively benign for continual learning, as the retention performance remains high and the transfer performance remains non-negative. However, the opposite, a combination of high feature similarity and low readout similarity is harmful. In this regime, both transfer and retention performance become below the chance level even when the two subsequent tasks are positively correlated. Furthermore, transfer performance depends on the feature similarity non-monotonically, such that, beyond a critical point, the higher the feature similarity is, the lower the transfer performance becomes.

We further analyze how common algorithms for continual learning, activity and plasticity gating [15, 42, 46], activity sparsification [57], and weight regularization [30, 36, 67], interact with task similarity in our problem setting, deriving several non-trivial conclusions. Activity gating improves retention at the cost of transfer when the gating highly sparsifies the activity, but helps both transfer and retention on average if the activity is kept relatively dense. Plasticity gating and activity sparsification, by contrast, do not influence either transfer or retention performance at the over-parameterized limit. Lastly, weight regularization in the Fisher information metric helps retention without affecting knowledge transfer and achieves perfect retention regardless of task similarity in the presence of low-dimensional latent. However, its diagonal approximation and the regularization in the Euclidean metric are much less robust against both task similarity and regularizer amplitude.

Furthermore, we test our key predictions numerically in a permuted MNIST task with a latent structure. When the input pixels are permuted from one task to the next, the retention performance remains high. However, when the mapping from the latent to the target output is changed, both the retention and transfer performance go below the chance level, as predicted. Random task-dependent gating of input and hidden layer activity improves retention at the cost of knowledge transfer, but adaptive gating mitigates this tradeoff. We also show that in a fully-connected feedforward network, there exists an efficient layer-wise approximation of the weight regularization in the Fisher information metric, which outperforms its diagonal approximation and the regularization in the Euclidean metric. Nevertheless, the performance of the diagonal approximation is much closer to the layer-wise approximation of the Fisher information metric than to the Euclidean weight regularization.

Our theory thus reveals when continual learning is difficult, and how different algorithms mitigate these challenging situations, providing a basic framework for analyzing continual learning in artificial and biological neural networks.

## 2 Related works

Previous works on continual learning in linear teacher-student models found that forgetting is most prominent at the intermediate task similarity [11, 40, 13] as observed empirically [47]. However, these works did not address the tradeoff between forgetting and knowledge transfer, and these simple settings did not disentangle the effect of the similarity in input feature and readout pattern. Forward and backward transfer performance in continual learning were also analyzed in linear and deep-linear networks [33, 8], yet its relationship with catastrophic forgetting has not been well characterized. Lee et al. [35] analyzed dynamics of both forgetting and forward transfer in one-hidden layer nonlinear network under a multi-head continual learning setting. However, their analysis of readout similarity

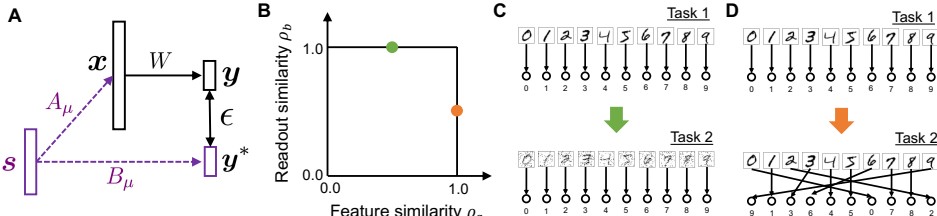

Figure 1: **(A)** Schematic representation of the continual linear regression model with low-dimensional latent variables. **(B-D)** Examples of continual learning with two tasks in the MNIST setting. The tasks have low feature similarity in panel C (input pixels are partially permuted) and low readout similarity in panel D (output labels are partially permuted). Panels C and D correspond to the green and orange points in panel B, respectively.

and its comparison to feature similarity were conducted numerically and it did not address the effect of common heuristics, such as gating or weight regularization, either.

By comparison, we introduce a low-dimensional latent structure into a linear teacher-student model, which enables us to decouple the influence of feature and readout similarity on knowledge transfer and retention. Moreover, this low-dimensionality assumption on the latent enables us to evaluate the transfer and retention performance analytically even in the presence of gating or weight regularization in the Fisher information metric.

Continual learning has been studied from many theoretical frameworks beyond teacher-student modeling, including neural tangent kernel [5, 9, 27], PAC learning [6, 60], and computational complexity [31]. Learning of low-dimensional latent representation has also been studied in the context of multi-task learning [43, 62]. In addition, several works investigated the effect of weight regularization on continual learning in analytically tractable settings [30, 12, 24]. In particular, Evron et al. [12] investigated effect of weight-regularization in Fisher-information metric in continual linear regression scenario.

## 3 Teacher-student model with low-dimensional latent variables

Let us consider task-incremental continual learning of regression tasks. For analytical tractability, we consider a teacher-student setting where the target outputs are generated by the teacher network (Fig. 1A). We define the student network, which learns the task, as a linear projection from a potentially nonlinear transformation of the input, $\boldsymbol{y} = W\psi(\boldsymbol{x})$, where $\boldsymbol{x} \in \mathbb{R}^{N_x}$ is the input, $\boldsymbol{y} \in \mathbb{R}^{N_y}$ is the output, $W \in \mathbb{R}^{N_y \times N_x}$ is the trainable weight matrix, and $\psi(\boldsymbol{x}) : \mathbb{R}^{N_x} \to \mathbb{R}^{N_x}$ is an input transformation. We use $\psi(\boldsymbol{x}) = \boldsymbol{x}$ for the vanilla and weight regularization models, $\psi(\boldsymbol{x}) = \boldsymbol{g} \odot \boldsymbol{x}$ for the task-dependent gating model, and $\psi(\boldsymbol{x}) = sgn(\boldsymbol{x}) \odot \max\{0, |\boldsymbol{x}| - \boldsymbol{h}\}$ for the soft-thresholding model, where $\boldsymbol{g}$ and $\boldsymbol{h}$ are gating and thresholding vectors, respectively. Here, $\boldsymbol{g} \odot \boldsymbol{x}$ is an element-wise multiplication, and $sgn(x)$ is a function that returns the sign of the input. Throughout the paper, we use bold-italic letters for vectors, capital-italic letters for matrices.

We generate the input $\boldsymbol{x}$ and target output $\boldsymbol{y}^*$ of the $\mu$-th task using a latent variable $\boldsymbol{s} \in \mathbb{R}^{N_s}$:

$$\boldsymbol{s} \leftarrow \mathcal{N}\left(0, I_s\right), \quad \boldsymbol{x} = A_\mu \boldsymbol{s}, \quad \boldsymbol{y}^* = B_\mu \boldsymbol{s}, \tag{1}$$

for $\mu = 1, 2$, where $A_\mu \in \mathbb{R}^{N_x \times N_s}$ and $B_\mu \in \mathbb{R}^{N_y \times N_s}$ are mixing matrices hidden from the student network, and $I_s$ is the size $N_s$ identity matrix. We introduce this latent structure, $\boldsymbol{s}$, to decouple the effect of feature and readout similarity on the transfer and retention performance. Below, we set the latent space to be low-dimensional compared to the input space (i.e., $N_s \ll N_x$). This is motivated by the presence of low-dimensional latent structure in many machine learning datasets [64, 7] and the tasks used in neuroscience experiments [17, 26], but also aids analytical tractability.

We generate the mixing matrices for the first task, $A_1$ and $B_1$, by sampling elements independently from a Gaussian distribution with mean zero and variance $\frac{1}{N_s}$. The subsequent task matrices $A_2$ and $B_2$ are also generated randomly, but with element-wise correlation with the previous matrices $A_1$ and $B_1$ (see Appendix A for the details). We denote the element-wise correlation between $A_1$ and

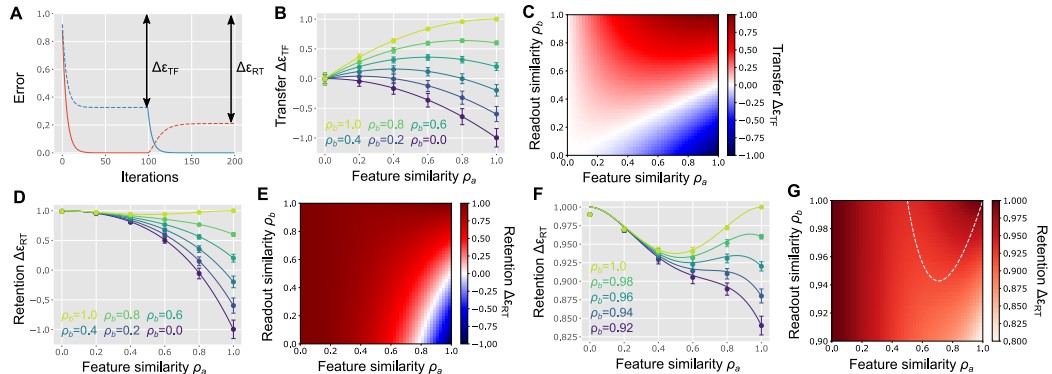

Figure 2: Transfer and retention performance of the vanilla model. **(A)** Illustration of $\Delta\epsilon_{TF}$ and $\Delta\epsilon_{RT}$. Red and blue lines represent the error on task 1 and task 2, respectively. Here, the model was trained on task 1 for 100 iterations and then trained on task 2 for another 100 iterations. **(B, C)** Transfer performance under various task similarity. Points in panel **B** are numerical results (the means and the standard deviations over ten random seeds), while solid lines are analytical results (Eq. 4). **(D-G)** Retention performance under various task similarity. Panel **G** magnifies the $0.9 \leq \rho_b \leq 1.0$ region of panel **E**, and the white dashed line in panel **G** represents local minima/maxima.

$A_2$ by $\rho_a$ and refer it as feature correlation, because $A_1$ and $A_2$ specify the input features. Similarly, we denote the correlation between $B_1$ and $B_2$, the readout correlation, by $\rho_b$. Figs. 1B-D illustrate the difference between $\rho_a$ and $\rho_b$ in an image classification setting. When $\rho_a < 1.0$ and $\rho_b = 1.0$, input features are partially modified in the second task compared to the first task (Green point on Fig. 1B and Fig. 1C). For example, permuted MNIST task [22] corresponds to this low feature similarity setting with $\rho_a = 0.0$ and $\rho_b = 1.0$. By contrast, when $\rho_a = 1$ and $\rho_b < 1$, the readouts are changed partially (Orange point on Fig. 1B and Fig. 1D). When $\rho_a = 1$ and $\rho_b = -1$, the same input features are associated with the opposite readout in the new task, which is known as the reversal learning paradigm. In this reversal scenario, the network fails to achieve knowledge transfer rather trivially. Thus, we focus on the scenario when tasks have non-negative correlation in terms of both input features and readout (i.e., $0 \leq \rho_a, \rho_b \leq 1$) below.

We measure the performance of the student network with weight $W$ on the $\mu$-th task by mean-squared error $\epsilon_\mu[W] \equiv \frac{1}{N_y} \left\langle \|B_\mu \boldsymbol{s} - W\psi(A_\mu \boldsymbol{s})\|^2 \right\rangle_{\boldsymbol{s}}$, where $\langle \cdot \rangle_{\boldsymbol{s}}$ is the expectation over latent variable $\boldsymbol{s}$. The transfer and retention performance are defined by

$$\Delta\epsilon_{TF} \equiv \langle \epsilon_2[W_o] - \epsilon_2[W_1] \rangle_{A,B}, \quad \Delta\epsilon_{RT} \equiv \langle \epsilon_1[W_o] - \epsilon_1[W_2] \rangle_{A,B}. \tag{2}$$

Here, $\langle \cdot \rangle_{A,B}$ is the expectation over randomly generated task matrices $A_1, A_2, B_1$ and $B_2$ under a given feature and readout correlation $\rho_a, \rho_b$. As illustrated in Fig. 2A, the transfer performance $\Delta\epsilon_{TF}$ measures how much performance the model achieves on task 2 by learning task 1, whereas the retention performance $\Delta\epsilon_{RT}$ measures how well the model can perform task 1 after learning task 2 (here, the task switch occurs at the 100th iteration). Below, we study how knowledge transfer and retention, the two key objectives of continual learning, depend on task similarity, and how to optimize the performance through gating and weight regularization.

## 4 Impact of task similarity on knowledge transfer and retention

Let us first investigate the vanilla model without gating or weight regularization, to examine how task similarity affects knowledge transfer and retention. Given $\psi(\boldsymbol{x}) = \boldsymbol{x}$, at the infinite sample limit, learning by gradient descent follows $\dot{W} = -(WA_\mu - B_\mu)A_\mu^T$. The fixed point of this dynamics, as detailed in Appendix B.1, is:

$$W_\mu = W_{\mu-1}(I - U_\mu U_\mu^T) + B_\mu A_\mu^+, \tag{3}$$

where $W_{\mu-1}$ is the weight after learning of the previous task, $U_\mu$ is the semi-orthogonal matrix from singular value decomposition (SVD) of $A_\mu = U_\mu \Lambda_\mu V_\mu^T$, and $A_\mu^+$ is the pseudo-inverse of matrix $A_\mu$.

Inserting Eq. 3 into Eq. 2 and taking the expectation over randomly generated tasks $A_1, B_1, A_2, B_2$, the transfer and retention performance are written as (see Appendix B.1)

$$\Delta\epsilon_{TF} = \rho_a(2\rho_b - \rho_a), \quad \Delta\epsilon_{RT} = 1 - \rho_a^2(\rho_a^2 - 2\rho_a\rho_b + 1). \tag{4}$$

Recall that $\rho_a$ is the feature similarity defined by the correlation between $A_1$ and $A_2$, while $\rho_b$ is the readout similarity, representing the correlation between $B_1$ and $B_2$. The derivation of the above equations relies on (correlated) random generation of $A_1, B_1, A_2, B_2$ and the low-rank latent assumption: $N_s \ll N_x$. These two equations, despite their simplicity, capture the transfer and retention performance in numerical simulations well (Figs. 2B, D, and F; see Appendix E.1 for the details of numerical estimation). Furthermore, they reveal asymmetric and non-monotonic impact of the feature and readout similarity on the performance.

Fig. 2B depicts the knowledge transfer from task 1 to task 2, $\Delta\epsilon_{TF}$, under various $(\rho_a, \rho_b)$ conditions. As expected, $\Delta\epsilon_{TF} = 0$ when the two tasks are independent (i.e., $(\rho_a, \rho_b) = (0, 0)$), and $\Delta\epsilon_{TF} = 1$ when the tasks are identical (i.e., $(\rho_a, \rho_b) = (1, 1)$). At intermediate levels of similarity, there is clear asymmetry in the influence of feature and readout similarities on transfer performance; particularly, a combination of high feature similarity and low readout similarity leads to negative transfer, while the opposite scenario results in a modest positive transfer (lower-right vs upper-left of Fig. 2B).

Notably, under a fixed readout similarity $\rho_b$, the knowledge transfer depends non-monotonically on the feature similarity $\rho_a$. When $\rho_a < \rho_b$, the higher the feature similarity the better transfer becomes, as implied from Eq. 4. However, once the feature similarity exceeds the readout similarity, counter-intuitively, high feature similarity worsens the transfer performance (Figs. 2B and C). This is because, when the feature similarity is high, inputs are aligned with learned features, resulting in a large output regardless of readout similarity. Particularly, under a low readout similarity, performance becomes below zero because extracted features are mostly projected to the incorrect directions.

The retention performance also exhibits asymmetric dependence on feature and readout similarity. When feature similarity $\rho_a$ is low, the network barely forgets the previous task regardless of readout similarity (Fig. 2D left). By contrast, when the feature similarity is high, the retention performance can either be positive or negative depending on the readout similarity. Moreover, in the high readout similarity regime, the retention performance is the lowest at an intermediate feature similarity (Figs. 2E and F). This is because high retention is possible when either interference is low, or similarity between two tasks is high. Note that, this last point on the non-monotonic dependence on feature similarity under $\rho_b = 1$ has been investigated both empirically [47] and analytically [35, 11, 13]. Indeed, at $\rho_b = 1$, $\Delta\epsilon_{RT}$ in Eq. 4 coincides with Eq. (5) in [13].

Therefore, the knowledge transfer and retention performance depend on feature and readout similarities in an asymmetric and non-monotonic manner. Specifically, a combination of high feature similarity and low readout similarity is detrimental to continual learning, resulting in negative transfer and retention performance, even in the presence of a positive correlation between the two tasks in both input features and readout patterns. Moreover, under a fixed readout similarity, the knowledge transfer performance depends on the feature similarity in a non-monotonic manner. Thus, the continual learning in the vanilla model is sensitive to task similarity and limited in performance. These results make us wonder whether we can mitigate the sensitivity to task similarity and improve the overall transfer and retention performance by modifying the learning algorithm. To this end, we next investigate how task-dependent gating and weight regularization methods, two popular strategies in continual learning, alleviate knowledge transfer and retention.

## 5 Task-dependent gating

### 5.1 Random activity gating

One popular method for mitigating forgetting in continual learning is activity gating [15, 54, 42, 21, 56]. With gating of input activity, the network is written as $\boldsymbol{y} = W(\boldsymbol{g}_\mu \odot \boldsymbol{x})$, where $\boldsymbol{g}_\mu \in \{0, 1\}^{N_x}$ is a binary gating vector for task $\mu$. We first consider a random gating scenario where elements of $\boldsymbol{g}_\mu$ are randomly sampled from a Bernoulli distribution with rate $\alpha$ which we denote as the gating level. All units are active at $\alpha = 1$, while no units are active at $\alpha \to 0$ limit.

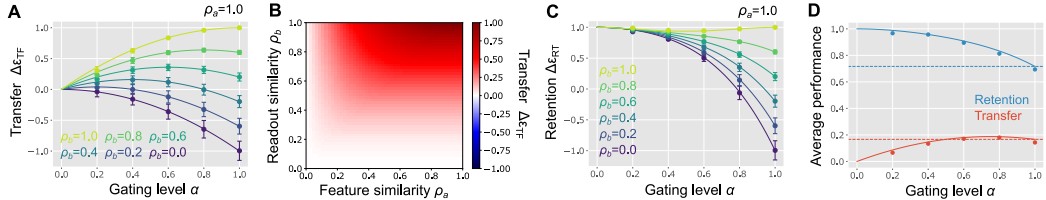

Figure 3: Random task-dependent activity gating model. **(A)** Knowledge transfer performance under $\rho_a = 1.0$. The gating level $\alpha$ is defined as the fraction of active input neurons (i.e., $\alpha = \Pr[g_i = 1]$). **(B)** The transfer performance under the optimal gating level $\alpha^* = \min\{\frac{\rho_b}{\rho_a}, 1\}$. **(C)** Retention performance under $\rho_a = 1.0$. **(D)** Average transfer and retention performance over uniform prior on $0 \leq \rho_a, \rho_b \leq 1$. Horizontal dashed lines are the performance of the vanilla model, while solid lines are the performance of the random gating model. Points are numerical estimations.

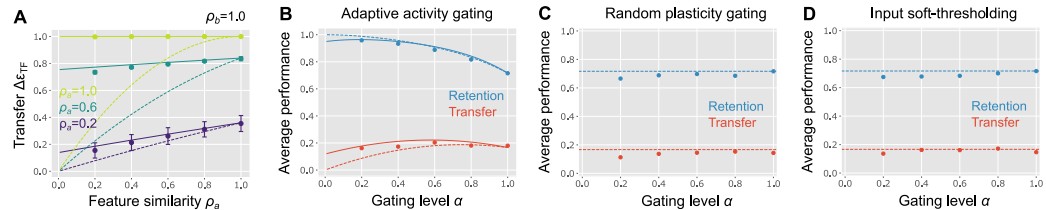

Figure 4: Transfer and retention performance of the adaptive activity gating (**A, B**), random plasticity gating (**C**), and input soft-thresholding (**D**) models. Dashed and solid lines in panels **A** and **B** represent the performance of the random and adaptive activity gating models, respectively.

From a parallel argument with the vanilla model, when $N_s \ll \alpha N_x$, the transfer and retention performance are estimated as (see Appendix B.1):

$$\Delta\epsilon_{TF} = \alpha\rho_a(2\rho_b - \alpha\rho_a), \tag{5a}$$

$$\Delta\epsilon_{RT} = 1 - \alpha^2\rho_a^2(\alpha^2\rho_a^2 - 2\alpha\rho_a\rho_b + 1). \tag{5b}$$

Thus, random gating scales the feature similarity from $\rho_a$ to $\alpha\rho_a$. This scaling lowers the knowledge transfer if $\rho_b \geq \rho_a$ because random gating reduces the fraction of input neurons active in two subsequent tasks (lime line in Fig. 3A). However, if $\rho_b < \rho_a$, gating with $\alpha \geq \frac{\rho_b}{\rho_a}$ enhances the transfer by reducing the effective feature similarity (blue lines in Fig. 3A; also compare Fig. 3B with Fig. 2C). The optimal gating level $\alpha^*$ for knowledge transfer is $\min\{\frac{\rho_b}{\rho_a}, 1\}$, indicating that the input activity typically needs to be dense. By contrast, the retention performance is optimized at $\alpha \to 0$ limit where tasks do not interfere with each others (Fig. 3C). At this limit, we have $\Delta\epsilon_{RT} \to 1$. Note that, in reality, $\alpha$ has to be non-zero to optimize the retention (Eq. 5b holds only when $\alpha \gg \frac{N_s}{N_x}$).

These results indicate that random activity gating improves the retention at the cost of forward knowledge transfer, as suggested previously [28, 39, 29]. This tradeoff between knowledge transfer and retention is especially critical when the network doesn't know the task similarities. Suppose that the task similarity is distributed uniformly over $0 \leq \rho_a, \rho_b \leq 1$. Then, the average transfer performance is maximized at $\alpha = \frac{3}{4}$, while the average retention performance decreases monotonically as a function of $\alpha$ (Fig. 3D). Thus, a moderate gating ($\alpha \approx \frac{3}{4}$) benefits both transfer and retention on average. However, retention performance in this regime is significantly below one, implying that the network cannot reliably achieve high retention performance without sacrificing the transfer performance.

## 5.2 Adaptive activity gating

One way to overcome the tradeoff between transfer and retention performance is to consider adaptive gating. Let us introduce a probe trial at the beginning of the task 2, in which the model tests how well it performs if it uses the gating for task 1, $\boldsymbol{g}_1$, for task 2 as well. If the error in the probe trial is small, the model should keep using the same gating vectors to achieve a good knowledge transfer. Otherwise, it should resample the gating vector for retaining the knowledge on task 1. Using the

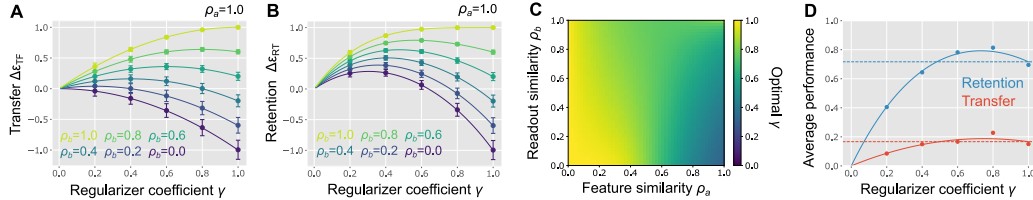

Figure 5: Performance of weight regularization in Euclidean metric. (**A,B**) Transfer (**A**) and retention (**B**) performance. The amplitude of the weight regularization scales with $\frac{1}{\gamma} - 1$. (**C**) Regularizer coefficient $\gamma$ that optimizes the retention performance. (**D**) Average performance over uniform task similarity distribution in $0 \leq \rho_a, \rho_b \leq 1$. Horizontal dashed lines are the performance of the vanilla model.

probe error $\epsilon_{pb}$, we set the probability of using the same gating as $\rho_g = 1 - \epsilon_{pb}/\epsilon_o$. This adaptive gating indeed improves the average transfer performance compared to the random gating (solid vs dashed lines in Fig. 4A) with only a relatively small reduction in the retention performance (Fig. 4B).

### 5.3 Plasticity gating and input soft-thresholding

Previous works have also explored other gating mechanisms, such as plasticity gating [46] and activity sparsification [57]. However, their benefits over the activity gating discussed above have not been fully understood. We implement task-dependent plasticity gating into our problem setting by making only the synaptic weights projected from a subset of input neurons plastic. Unexpectedly, we found that both transfer and retention performance are independent of the fraction of plastic synapses in the $N_s \ll N_x$ limit, potentially due to over-parameterization (Fig. 4C; see Appendix B.2 for details).

Similarly, when the input activity is sparsified using soft-thresholding $\varphi(x) = sgn(x) \max\{0, |x| - h\}$, with a fixed threshold $h = \sqrt{2}\text{erfc}^{-1}(\alpha)$, the transfer and retention performance remain independent of the resultant input sparsity $\alpha$ (Fig. 4D; see Appendix B.3 for the details). These results imply that plasticity gating and input sparsification are not effective in our model, which operates in an over-parameterized regime (i.e., $N_s \ll N_x$), but do not exclude their potential benefits for many continual learning tasks.

## 6 Weight regularization

### 6.1 Weight regularization in Euclidean metric

Another popular method is weight regularization, which keeps the synaptic weights close to those learned from previous tasks [30, 36, 67]. Let us first consider regularization of the Euclidean distance between the current weight $W$ and the weight learned in the previous task $W_{\mu-1}$:

$$\ell_\mu = \frac{1}{2} \|B_\mu - W A_\mu\|_F^2 + \frac{N_x}{2N_s}\left(\frac{1}{\gamma} - 1\right)\|W - W_{\mu-1}\|_F^2. \tag{6}$$

Here, we parameterize the regularizer amplitude by $\frac{N_x}{N_s}\left(\frac{1}{\gamma} - 1\right)$ using a non-negative parameter $\gamma$ for brevity. $\gamma = 1$ corresponds to zero regularization, while $\gamma = 0$ is the infinite regularization limit. Under this parameterization, if $N_s \ll N_x$, the transfer and retention performance follow simple expressions as below (see Appendix C.1):

$$\Delta\epsilon_{TF} = \gamma\rho_a(2\rho_b - \gamma\rho_a), \tag{7a}$$

$$\Delta\epsilon_{RT} = 1 - \gamma^2\rho_a^2(1 - 2\gamma\rho_a\rho_b + \gamma^2\rho_a^2) + 2\gamma\rho_a(1 - \gamma)(\rho_b - \gamma\rho_a) - (1 - \gamma)^2. \tag{7b}$$

The expression of the transfer performance is equivalent to Eq. 5a if we change $\gamma$ to $\alpha$. Thus, the Euclidean weight regularization with amplitude $\frac{N_x}{N_s}\left(\frac{1}{\gamma} - 1\right)$ is mathematically equivalent with random activity gating with sparsity $\gamma$, in terms of knowledge transfer (compare Fig. 5A with Fig. 3A). By contrast, the expression of the retention performance contains two additional terms compared to 5b. In particular, the last term, $(1 - \gamma)^2$, indicates that strong weight regularization not only prevent forgetting, but also impairs task acquisition. Thus, the retention performance is optimized at an intermediate regularizer strength (Figs. 5B and C).

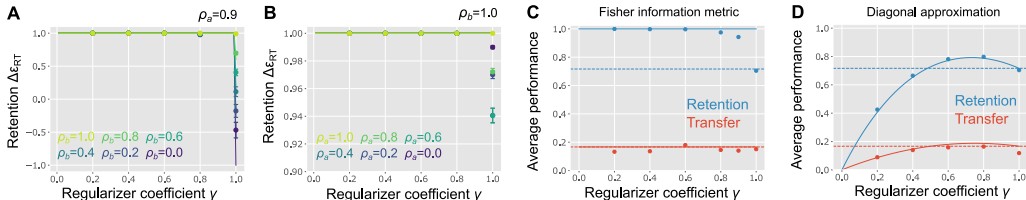

Figure 6: Weight regularization in the Fisher information metric. (**A,B**) The retention performance under various task similarities and regularizer coefficients. (**C,D**) Average transfer and retention performance under the regularization with the exact Fisher information metric (**C**) and its diagonal approximation (**D**).

Because strong regularization (i.e., small $\gamma$) impairs both retention and transfer, unlike the random activity gating model, there isn't a tradeoff between knowledge transfer and retention in terms of the average performance over $0 \leq \rho_a, \rho_b \leq 1$ (Fig. 5D). However, the retention performance at the optimal parametric regime is relatively low (Fig. 5D with Fig. 4B).

## 6.2  Weight regularization in Fisher information metric

Previous works proposed that the synaptic weights should be regularized in the Fisher information metric to allow flexibility in the non-overlapping weight space [30, 67]. Applying it to our model setting, the regularizer term in Eq. 6 instead becomes $\frac{1}{2}(\frac{1}{\gamma}-1)\left\|(W-W_{\mu-1})A_{\mu-1}\right\|_F^2$ (see Appendix C.2). If $\rho_a, \gamma < 1$ and $N_s \ll N_x$, optimization of the weight under this regularization yields

$$W_\mu = W_{\mu-1}\left(I - \frac{N_s}{N_x(1-\rho_a^2)}A_\mu\left[A_\mu^T - \rho_a A_{\mu-1}^T\right]\right) + \frac{N_s}{N_x(1-\rho_a^2)}B_\mu\left(A_\mu^T - \rho_a A_{\mu-1}^T\right), \quad (8)$$

where $W_{\mu-1}$ is the weight after the previous task learning. Notably, the weight becomes independent of the regularizer's amplitude $\gamma$. Furthermore, the retention performance under arbitrary task similarity $(\rho_a, \rho_b)$ is derived as $\Delta\epsilon_{RT} = 1 - \mathcal{O}(\frac{N_s}{N_x(1-\rho_a^2)})$. Thus, under the Fisher information metric, the retention performance is perfect as long as the condition $N_s \ll N_x(1-\rho_a^2)$ is satisfied (Figs. 6A-C). Intuitively, assuming over-parameterization ($N_s \ll N_x$), the network can freeze the weight changes in a low-dimensional subspace, while maintaining sufficient plasticity for new tasks, unless the two tasks share the same feature. Notably, if the first task is learned with weight regularization in an orthogonal direction, the transfer performance remains the same with the vanilla model (Fig. 6C).

Importantly, this invariance no longer holds when the Fisher information metric is approximated by its diagonal component (Figs. 6D vs 6C), as is done in the elastic weight methods [30]. This is because the diagonal approximation makes the metric full-rank even when the true metric has a low-rank structure. Thus, weight regularization in the Fisher information metric robustly helps retention without harming transfer, but diagonal approximation attenuates its robustness.

## 7  Numerical experiments

Our theory so far has revealed when continual learning is difficult and when activity gating and weight regularization can ameliorate the transfer and retention in a simple problem setting. Let us next examine how much these insights are applicable to more realistic datasets and neural architectures.

To this end, we consider the permuted MNIST dataset [34, 22], but with a latent structure (see Appendix E.2). For each output label, we constructed a four-dimensional latent variable $s$ using the binary representation of the corresponding digit (e.g., $s_9 = [1, 0, 0, 1]^T$). The target output was then generated by a random fixed projection of the latent variable: $y^* = B_\mu(s - \frac{1}{2})$. Note that, although here we introduced an explicit latent structure for a direct comparison with the theory, qualitatively similar results holds in the vanilla permuted MNIST setting depicted in Figs. 1B-D (see Fig. 12). We controlled the readout similarity between tasks using the element-wise correlation between two matrices $B_1$ and $B_2$. The feature similarity was controlled by permuting a subset of input pixels, as in the previous permuted MNIST tasks. We used a one-hidden-layer fully-connected network with rectified-linear nonlinearity at the hidden layer, and trained the network using stochastic gradient descent.

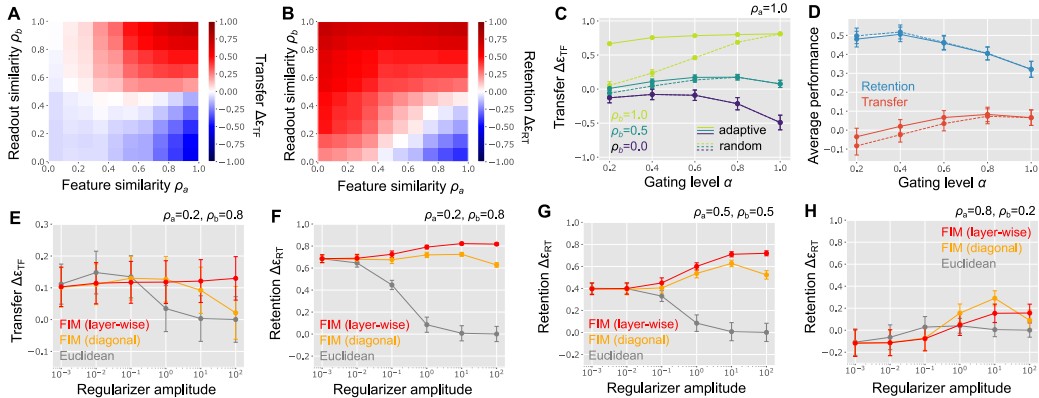

Figure 7: Permuted MNIST with latent variables. (**A,B**) Transfer and retention performance of the vanilla model. (**C,D**) Performance of random (dashed lines) and adaptive (solid lines) activity gating models. (**E-H**) Performance of the weight regularization in the Euclidean metric, and approximated Fisher information metrics (red: layer-wise approximation; orange: synapse-wise/diagonal approximation). Here, lines are linear interpolations and error bars are standard errors over random seeds, not standard deviations.

As predicted from our theory, both transfer and retention performance, measured by the test error, exhibited asymmetric and non-monotonic dependence on the task similarity. When $(\rho_a, \rho_b) = (1.0, 0.0)$, both transfer and retention performance were below zero (lower-right of Figs. 7A and B vs. Figs. 2C and E). By contrast, the model achieved high retention and zero transfer under $(\rho_a, \rho_b) = (0.0, 1.0)$ as expected (upper-left). Notably, around $\rho_b = 0.5$, the transfer performance exhibited a non-monotonic dependence on $\rho_a$ in accordance with the theory. The retention performance also exhibited non-monotonic dependence on $\rho_a$ under $\rho_b \approx 1$, but the dependence became monotonic after a long training (see Figs. 9C and D in Appendix). We observe consistent results even when feature and readout similarity are adjusted by permuting input pixels and output labels, respectively, provided the loss is measured by cross-entropy (Fig. 12B, corresponding to the examples depicted in Figs. 1C and 1D).

Random activity gating, implemented in both input and hidden layers, improved the retention performance at the cost of low knowledge transfer as predicted (compare Fig. 7D and Fig. 4B). Moreover, adaptive gating based on a probe trial improved the transfer performance under a low gating level, especially when the readout similarity is high (solid vs dashed lines in Figs. 7C and D).

In a deep neural network, weight regularization using the Fisher information metric is typically computationally expensive. However, an efficient layer-wise approximation exists for fully-connected networks (see Appendix E.2). This layer-wise approximation enabled high retention performance across a wide range of regularization amplitudes and consistently outperformed weight regularization in the Euclidean metric under various task similarities (red vs. gray lines in Figs. 7F-H). Nevertheless, the retention performance under the diagonal approximation was closer to that of the layer-wise approximation of the Fisher information metric and was slightly better when both methods performed poorly (red vs. orange lines). This is because the sparsity of hidden layer weights and activity makes the diagonal components sparse.

## 8 Discussion

Here, we analyzed the impact of task similarity on continual learning in a linear teacher-student model with low-dimensional latent structures. We showed that, a combination of high feature similarity and low readout similarity lead to poor outcomes in both retention and transfer, unlike the opposite combination. We further explored how continual learning algorithms such as gating mechanisms, activity sparsification, and weight regularization interact with task similarity. Results indicate that weight regularization in the Fisher information metric significantly aids retention regardless of task similarity. Numerical experiments in a permuted MNIST task with latent supported these findings.

Our findings on the interaction between task similarity and continual learning algorithms have several implications. Firstly, when tasks are known to have high feature similarity and low readout similarity, adaptive activity gating and weight regularization in the Fisher information metric help retention without sacrificing transfer (Figs. 4, 6, 7). In the context of neuroscience, our results indicate that simple non-adaptive activity or plasticity gating mechanisms are not sufficient for good continual learning performance, especially when familiar sensory stimuli need to be associated with novel motor actions. Indeed, a previous study reported catastrophic forgetting among rats learning timing estimation tasks [14]. Lastly, it also implies the potential importance of studying wider benchmarks beyond permuted image recognition tasks for continual learning, because image permutation belongs to a class of relatively harmless task similarity according to our theory. As a simple extension, here we developed an image recognition task with a latent variable and showed that a vanilla deep network exhibits more forgetting under readout remapping than under input permutation (Figs. 7A and B).

**Limitations**  Our theoretical results from a teacher-student model come with limitation in direct applicability. The presence of low-dimensional latent generating both input and target output is a reasonable assumption for many real-world tasks [7, 26], but the linear projection assumption is not. While we replicated most key results in a deep nonlinear network solving permuted MNIST task, we found that the diagonal approximation of Fisher information metric performs much better than the prediction from the teacher-student model. This is potentially because sparse activity at the hidden layer effectively makes the regularization low-rank even under the diagonal approximation. More generally, the presence of hidden layers should enable a distinctive adaptation for feature and readout similarity, which is an important future direction. It is also important to apply our theoretical framework for analyzing continual learning in more complicated neural architectures and datasets.

# 9  Acknowledgments

The author thanks Liu Ziyin and Ziyan Li for discussion. This work was partially supported by The Swartz Foundation.

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

# A   Problem setting: Linear teacher-student model with a latent variable

Let us consider a continuous learning of $N_{sess}$ tasks. We generate the input $\boldsymbol{x} \in \mathbb{R}^{N_x}$ and the target output $\boldsymbol{y}^* \in \mathbb{R}^{N_y}$ of the $\mu$-th task by

$$\boldsymbol{s} \leftarrow \mathcal{N}\left(0, I_s\right), \quad \boldsymbol{x} = A_\mu \boldsymbol{s}, \quad \boldsymbol{y}^* = B_\mu \boldsymbol{s}, \tag{9}$$

where $\boldsymbol{s} \in \mathbb{R}^{N_s}$ is the latent variable, $I_s$ is the size $N_s$ identity matrix, and $A_\mu \in \mathbb{R}^{N_x \times N_s}$ and $B_\mu \in \mathbb{R}^{N_y \times N_s}$ are mixing matrices that map the latent variable $\boldsymbol{s}$ into the input space and output space, respectively. Throughout the paper, the vector $\boldsymbol{x}$ is defined as a column vector, and its transpose $\boldsymbol{x}^T$ is a row vector.

We consider sequential learning of these tasks with a neural network defined by $\boldsymbol{y} = W\psi(\boldsymbol{x})$, where $W \in \mathbb{R}^{N_y \times N_x}$ is the plastic weight. We set the function $\psi : \mathbb{R}^{N_x} \to \mathbb{R}^{N_x}$ to $\psi(\boldsymbol{x}) = \boldsymbol{x}$ in the vanilla and weight regularization models, $\psi(\boldsymbol{x}) = \boldsymbol{g} \odot \boldsymbol{x}$ in activity and plasticity gating models, and $\psi(\boldsymbol{x}) = sgn(\boldsymbol{x}) \odot \max\{0, |\boldsymbol{x}| - \boldsymbol{h}\}$ in the soft-thresholding model. Here, $\boldsymbol{g} \odot \boldsymbol{x}$ represents an element-wise multiplication of two vectors, and $sgn(x)$ is a function that returns the sign of the input $x$. The goal of the student network, $\boldsymbol{y} = W\psi(\boldsymbol{x})$, is to mimic the teacher network that generates both input and target output [18, 66, 3]. Below, we focus on learning of $N_{sess} = 2$ tasks. To analyze the transfer and retention performance, we introduce the following two assumptions on task structure.

**Assumption I: Random task assumption**   We consider random generation of task matrices $A_\mu, B_\mu$ but with correlation between subsequent tasks. We sample elements of the mixing matrices for the first task, $A_1$ and $B_1$, from a Gaussian distribution with mean zero and variance $\frac{1}{N_s}$ independently. For the subsequent tasks, we generate mixing matrices $A_\mu$ by

$$A_{\mu,ij} = \begin{cases} A_{\mu-1,ij} & \text{(with probability } \rho_a) \\ \widetilde{A}_{\mu,ij} & \text{(otherwise)} \end{cases} \tag{10}$$

where $\widetilde{A}_\mu$ is a random matrix whose elements are sampled independently from a Gaussian distribution with mean zero and variance $\frac{1}{N_s}$. We generate $B_\mu$ in the same manner, but with correlation $\rho_b$. Although this hidden weight generation doesn't exactly follow the correlated Gaussian assumption, the difference appears only in higher-order terms, which are negligible under $N_s \gg N_x$ assumption we introduce below.

**Assumption II: Low-dimensional latent assumption**   The second assumption is the relative low dimensionality of the latent space with respect to the input space: $N_s \ll N_x$. This assumption of over-parameterization in the input space simplifies the analysis (see Appendix D).

**Evaluation of the transfer and retention performance**   We evaluate the task performance of a student model with weight $W$ on task $\mu$ by mean-squared error:

$$\epsilon_\mu[W] \equiv \frac{1}{N_y} \left\langle \|B_\mu \boldsymbol{s} - W\psi(A_\mu \boldsymbol{s})]\|^2 \right\rangle_{\boldsymbol{s}}. \tag{11}$$

Here, $\langle \cdot \rangle_{\boldsymbol{s}}$ is the expectation over latent $\boldsymbol{s} \sim \mathcal{N}(0, I_s)$. Using $\epsilon_\mu[W]$, the degree of forward knowledge transfer and retention are evaluated by

$$\Delta\epsilon_\mu^{TF} \equiv \langle \epsilon_\mu[W_o] - \epsilon_\mu[W_{\mu-1}] \rangle_{A,B}, \tag{12a}$$

$$\Delta\epsilon_\mu^{RT} \equiv \langle \epsilon_\mu[W_o] - \epsilon_\mu[W_{\mu+1}] \rangle_{A,B}, \tag{12b}$$

where $W_\mu$ is the weight after training on the $\mu$-th task. Note that, the retention performance is defined by the difference with the initial error $\epsilon_\mu[W_o]$. This means that, if the network fails to learn the task in the first place (i.e., if $\epsilon_\mu[W_\mu] > 0$), the retention performance becomes sub-optimal even if the network doesn't forget the learned task. We used this definition for our problem setting because the network is able to learn the task perfectly unless strong regularization is added to the network.

**Learning procedures**   We consider the gradient descent learning from infinite samples at the gradient flow limit and analyze the performance at the convergence. Below, we investigate how $\Delta\epsilon_\mu^{TF}$ and $\Delta\epsilon_\mu^{RT}$ depend on the task similarity and hyper-parameters under various algorithms for continual learning.

# B Task-dependent gating model

## B.1 Activity gating

Let us first consider task-dependent activity gating model. The student network (the network that learns the task) is given by:

$$\boldsymbol{y} = W[\boldsymbol{g}_\mu \odot \boldsymbol{x}], \tag{13}$$

where $\boldsymbol{g}_\mu$ is the gating vector that depends on the task id $\mu$, but independent of input $\boldsymbol{x}$. Note that, by setting $\boldsymbol{g}_\mu = \mathbf{1}$, we recover the vanilla model $\boldsymbol{y} = W\boldsymbol{x}$.

The performance of this network on the $\mu$-th task is written as

$$\epsilon_\mu[W] \equiv \frac{1}{N_y} \left\langle \|B_\mu \boldsymbol{s} - W[\boldsymbol{g}_\mu \odot (A_\mu \boldsymbol{s})]\|^2 \right\rangle_{\boldsymbol{s}} = \frac{1}{N_y} \|B_\mu - W D_\mu A_\mu\|_F^2, \tag{14}$$

where $D_\mu \equiv \mathrm{diag}[\boldsymbol{g}_\mu]$ is a diagonal matrix whose $(i,i)$-th element is $g_{\mu,i}$, and $\|\cdot\|_F$ is the Frobenius norm.

**Solution of gradient descent learning**   Under the gradient descent learning, the weight dynamics follows

$$\dot{W}(t) = -\eta \frac{\partial \epsilon_\mu[W]}{\partial W} = -\frac{2\eta}{N_y} \left( W D_\mu A_\mu - B_\mu \right) \left( D_\mu A_\mu \right)^T. \tag{15}$$

From singular value decomposition (SVD), $D_\mu A_\mu$ is rewritten as

$$D_\mu A_\mu = U_\mu \Sigma_\mu V_\mu^T, \tag{16}$$

where $U_\mu \in \mathbb{R}^{N_x \times N_o}$ and $V_\mu \in \mathbb{R}^{N_s \times N_o}$ are semi-orthonormal matrices (i.e., $U_\mu^T U_\mu = V_\mu^T V_\mu = I_o$), $\Sigma_\mu \in \mathbb{R}^{N_o \times N_o}$ is a positive diagonal matrix, $N_o$ is the number of non-zero singular values of $D_\mu A_\mu$. Using this decomposition, the gradient descent dynamics is rewritten as

$$\dot{W}(t) = -\frac{2\eta}{N_y} \left( W U_\mu \Sigma_\mu - B_\mu V_\mu \right) \Sigma_\mu U_\mu^T. \tag{17}$$

Thus, the weight matrix $W(t)$ at arbitrary $t$ is written as

$$W(t) = W(t=0) + Q(t)U_\mu^T, \tag{18}$$

where $Q \in \mathbb{R}^{N_y \times N_o}$ is a time-dependent matrix. At the fixed point of this learning dynamics, we have

$$\left( \left[W_{\mu-1} + QU_\mu^T\right] U_\mu \Sigma_\mu - B_\mu V_\mu \right) \Sigma_\mu U_\mu^T = O, \tag{19}$$

where $O$ is the zero matrix and $W_{\mu-1} \equiv W(t=0)$ is the weight after learning of the previous task. Because $\Sigma_\mu$ is a positive diagonal matrix, the equation above has a unique solution:

$$Q = \left( B_\mu V_\mu - W_{\mu-1} U_\mu \Sigma_\mu \right) \Sigma_\mu^{-1}, \tag{20a}$$

$$W = W_{\mu-1} \left( I_x - U_\mu U_\mu^T \right) + B_\mu (D_\mu A_\mu)^+, \tag{20b}$$

where $(D_\mu A_\mu)^+ = V_\mu \Sigma_\mu^{-1} U_\mu^T$ is the pseudo-inverse of $D_\mu A_\mu$, and $I_x$ is the size $N_x$ identity matrix. From Eqs. 20b and 14, we have

$$\epsilon_\mu[W_\mu] = \frac{1}{N_y} \left\| B_\mu \left( V_\mu V_\mu^T - I_s \right) \right\|_F^2. \tag{21}$$

Moreover, by setting the initial weight $W_o$ (the weight before the first tasks) to zero, we have

$$\epsilon_\mu[W_o] = \frac{1}{N_y} \|B_\mu\|_F^2, \tag{22}$$

and $W_1$ and $W_2$ follow

$$W_1 = B_1(D_1 A_1)^+, \quad W_2 = B_1(D_1 A_1)^+(I_x - U_2 U_2^T) + B_2(D_2 A_2)^+. \tag{23}$$

The error in task 2 after learning of task 1 is thus written as

$$\epsilon_2[W_1] = \frac{1}{N_y} \left\| B_2 - B_1 (D_1 A_1)^+ D_2 A_2 \right\|_F^2 . \tag{24}$$

Similarly, the error on task 1 after learning of task 2 is

$$\epsilon_1[W_2] = \frac{1}{N_y} \left\| B_1 \left( I_s - (D_1 A_1)^+ (I_x - U_2 U_2^T) D_1 A_1 \right) - B_2 (D_2 A_2)^+ D_1 A_1 \right\|_F^2 \tag{25}$$

Note that the results so far does not rely on Assumptions I and II, making them applicable to arbitrary task matrices $(A, B)$.

The weight matrix after learning depends on the pseudo-inverse, $(DA)^+ = V \Sigma^{-1} U^T$, which is typically a complicated function of the original matrix $DA$. However, when $\alpha N_x \gg N_s$, it can be approximated by a scaled transpose, $\frac{N_s}{\alpha N_x} (DA)^T$ (see Eq. 101).

**Random and adaptive activity gating**  In the adaptive activity gating model, we generate input gating units $\{g_i^\mu\}$ randomly, but with correlations between subsequent tasks. We sample the gating units for the first task with $g_i^1 \leftarrow \text{Bernoulli}(\alpha)$, where $\alpha$ is the sparsity of the gating units. The gating units for the $\mu + 1$ task, $\{g_i^{\mu+1}\}$, are then generated by

$$g_i^{\mu+1} = \begin{cases} g_i^\mu & \text{(with probability } \rho_{\mu+1}^g) \\ \text{Bernoulli}(\alpha) & \text{(otherwise)} \end{cases} \tag{26}$$

Here, $\rho_{\mu+1}^g$ is the correlation between the gating units for the $\mu$-th and $\mu + 1$-th tasks. The gating correlation $\rho_\mu^g$ is set to zero in the case of random task-dependent gating, whereas $\rho_\mu^g$ is adjusted based on the model performance right after a model switch in the adaptive task-dependent gating. To this end, we introduce a probe trial where the model solves the new task using the gating vector for the old task. The error on the probe trial is

$$\epsilon_{pb} \equiv \frac{1}{N_y} \left\| B_2 - W_1 D_1 A_2 \right\|_F^2 . \tag{27}$$

We then set the probability of using the same gating elements by

$$\rho_g = \max \left\{ 0, 1 - \frac{\epsilon_{pb}}{\epsilon_o} \right\} \tag{28}$$

where $\epsilon_o \equiv \frac{1}{N_y} \left\| B_2 \right\|_F^2$ is the baseline error on the task 2. It keeps the gating the same if $\epsilon_{pb} = 0$, while it resamples all gating elements in case $\epsilon_{pb} \geq \epsilon_o$.

**Transfer performance**  The average transfer performance $\Delta \epsilon_{TF}$ from the first to the second tasks is:

$$\Delta \epsilon_{TF} = \frac{1}{N_y} \left\langle \| B_2 \|_F^2 \right\rangle - \frac{1}{N_y} \left\langle \left\| B_2 - B_1 (D_1 A_1)^+ D_2 A_2 \right\|_F^2 \right\rangle$$

$$= \frac{1}{N_y} \left\langle 2\text{tr} \left[ B_2^T B_1 (D_1 A_1)^+ D_2 A_2 \right] - \left\| B_1 (D_1 A_1)^+ D_2 A_2 \right\|_F^2 \right\rangle$$

$$= \frac{2\rho_b}{N_s} \left\langle \text{tr} \left[ (D_1 A_1)^+ D_2 A_2 \right] \right\rangle - \frac{1}{N_s} \left\langle \left\| (D_1 A_1)^+ D_2 A_2 \right\|_F^2 \right\rangle . \tag{29}$$

In the last line, we took the expectation over the correlated random matrices $B_1$ and $B_2$. Using the approximation $(DA)^+ \approx \frac{N_s}{\alpha N_x} (DA)^T$ (Eq. 101), and then taking the expectation over $A_1$, $A_2$, $D_1$, and $D_2$, the first term becomes

$$\left\langle \text{tr}[(D_1 A_1)^+ D_2 A_2] \right\rangle \approx \frac{N_s}{\alpha N_x} \left\langle \text{tr}[A_1^T D_1 D_2 A_2] \right\rangle$$

$$= \frac{N_s}{\alpha N_x} \sum_{i=1}^{N_s} \sum_{j=1}^{N_x} \left\langle a_{ji}^{(1)} g_j^{(1)} g_j^{(2)} a_{ji}^{(2)} \right\rangle = \tilde{\alpha} \rho_a N_s, \tag{30}$$

where $a_{ji}^{(1)}$ represents the $(j, i)$-th element of $A_1$, and $\tilde{\alpha}$ is defined by

$$\tilde{\alpha} \equiv \rho_g + (1 - \rho_g) \alpha. \tag{31}$$

The second term of Eq. 29 follows

$$\left\langle \left\| (D_1 A_1)^+ D_2 A_2 \right\|_F^2 \right\rangle \approx \left( \tfrac{N_s}{\alpha N_x} \right)^2 \left\langle \left\| A_1^T D_1 D_2 A_2 \right\|_F^2 \right\rangle$$

$$= \left( \tfrac{N_s}{\alpha N_x} \right)^2 \sum_{i,k=1}^{N_s} \sum_{j,l=1}^{N_x} \left\langle g_j^{(1)} g_j^{(2)} g_l^{(1)} g_l^{(2)} a_{ji}^{(1)} a_{jk}^{(2)} a_{lk}^{(2)} a_{li}^{(1)} \right\rangle$$

$$= \frac{1}{\alpha^2 N_x^2} \sum_{i,k=1}^{N_s} \sum_{j,l=1}^{N_x} \left\langle g_j^{(1)} g_j^{(2)} g_l^{(1)} g_l^{(2)} [\delta_{ik} \rho_a^2 + \delta_{jl} + \delta_{ik} \delta_{jl} \rho_a^2] \right\rangle$$

$$= N_s \left( \tilde{\alpha}^2 \rho_a^2 + \frac{\tilde{\alpha} N_s}{\alpha N_x} \left( 1 + \frac{\rho_a^2}{N_s} \right) \right). \tag{32}$$

The third line follows from Isserlis' theorem, from which we can decompose the higher-order correlation as below:

$$\left\langle a_{ji}^{(1)} a_{jk}^{(2)} a_{lk}^{(2)} a_{li}^{(1)} \right\rangle = \left\langle a_{ji}^{(1)} a_{jk}^{(2)} \right\rangle \left\langle a_{lk}^{(2)} a_{li}^{(1)} \right\rangle + \left\langle a_{ji}^{(1)} a_{li}^{(1)} \right\rangle \left\langle a_{jk}^{(2)} a_{lk}^{(2)} \right\rangle + \left\langle a_{ji}^{(1)} a_{lk}^{(2)} \right\rangle \left\langle a_{jk}^{(2)} a_{li}^{(1)} \right\rangle$$

$$= \frac{\rho_a^2}{N_s^2} \delta_{ik} + \frac{1}{N_s^2} \delta_{jl} + \frac{\rho_a^2}{N_s^2} \delta_{ik} \delta_{jl}. \tag{33}$$

Summing up the equations above, at $\frac{N_s}{\alpha N_x} \to 0$ limit, we have

$$\Delta \epsilon_{TF} = \tilde{\alpha} \rho_a \left( 2\rho_b - \tilde{\alpha} \rho_a \right). \tag{34}$$

Note that when $A_2$ is generated by random replacement, instead of sampling from a correlated Gaussian distribution, $\left\langle \left\| (D_1 a_1)^+ D_2 A_2 \right\|_F^2 \right\rangle$ term instead follows $\left\langle \left\| (D_1 a_1)^+ D_2 A_2 \right\|_F^2 \right\rangle \approx N_s \left( \tilde{\alpha}^2 \rho_a^2 + \frac{\tilde{\alpha} N_s}{\alpha N_x} \left[ 1 + \frac{\rho_a}{N_s} (2 - \rho_a \alpha \tilde{\alpha}) \right] \right)$, which converges to Eq. 32 at $N_s \ll N_x$ limit.

**Retention performance** Let us next analyze the retention performance, $\Delta \epsilon_{RT} \equiv \left\langle \epsilon_1[W_o] - \epsilon_1[W_2] \right\rangle_{A,B}$, which characterizes how well the network performs the task 1 after learning task 2. At $\alpha N_x \gg N_s$ regime, $\left\langle \epsilon_1[W_o] \right\rangle_{A,B} = 1$. Inserting Eq. 23 into Eq. 14, we get

$$\epsilon_1[W_2] = \frac{1}{N_y} \left\| B_1 - \left[ B_1 (D_1 A_1)^+ (I_x - U_2 U_2^T) + B_2 (D_2 A_2)^+ \right] D_1 A_1 \right\|_F^2$$

$$= \frac{1}{N_y} \left\| B_1 (D_1 A_1)^+ U_2 U_2^T D_1 A_1 - B_2 (D_2 A_2)^+ D_1 A_1 \right\|_F^2. \tag{35}$$

Thus, $\Delta \epsilon_{RT}$ follows

$$\Delta \epsilon_{RT} = \frac{1}{N_y} \left\langle \left\| B_1 \right\|_F^2 \right\rangle - \frac{1}{N_y} \left\langle \left\| B_1 (D_1 A_1)^+ U_2 U_2^T D_1 A_1 \right\|_F^2 \right\rangle - \frac{1}{N_y} \left\langle \left\| B_2 (D_2 A_2)^+ D_1 A_1 \right\|_F^2 \right\rangle$$

$$+ \frac{2}{N_y} \left\langle \mathrm{tr} \left[ B_2^T B_1 (D_1 A_1)^+ U_2 U_2^T D_1 A_1 (D_1 A_1)^T \left( (D_2 A_2)^+ \right)^T \right] \right\rangle$$

$$= 1 - \frac{1}{N_s} \left\langle \left\| (D_1 A_1)^+ U_2 U_2^T D_1 A_1 \right\|_F^2 \right\rangle - \frac{1}{N_s} \left\langle \left\| (D_2 A_2)^+ D_1 A_1 \right\|_F^2 \right\rangle$$

$$+ \frac{2\rho_b}{N_s} \left\langle \mathrm{tr} \left[ (D_1 A_1)^+ U_2 U_2^T D_1 A_1 (D_1 A_1)^T \left( (D_2 A_2)^+ \right)^T \right] \right\rangle. \tag{36}$$

Because $D_2 A_2 A_2^T D_2 = U_2 \Sigma_2^2 U_2^T$, at the $N_s \ll \alpha N_x$ limit, $U_2 U_2^T$ is approximated by (see Eq. 102)

$$U_2 U_2^T \approx \frac{N_s}{\alpha N_x} D_2 A_2 A_2^T D_2. \tag{37}$$

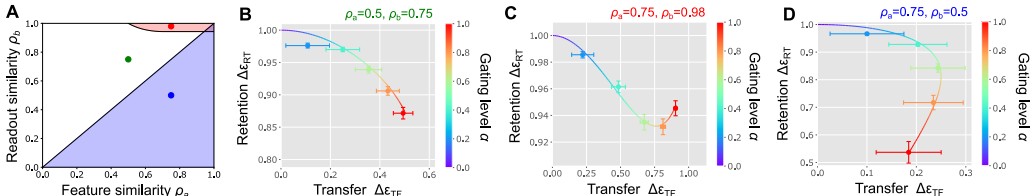

Figure 8: The gating level dependence of the transfer and retention performance (**A**) Phase diagram of the gating level dependence. (**B-D**) Transfer and retention performance as a function of the gating level at a representative point of each phase.

Using this approximation and $(DA)^+ \approx \frac{N_s}{\alpha N_x}(DA)^T$ (Eq. 101), we can obtain an analytical expression for Eq. 36. Taking the expectation over $D$ and $A$, the second term of Eq. 36 becomes

$$\frac{1}{N_s}\left(\frac{N_s}{\alpha N_x}\right)^4 \left\langle \left\| A_1^T D_1 D_2 A_2 A_2^T D_2 D_1 A_1 \right\|_F^2 \right\rangle$$

$$= \sum_{ijkl}\sum_{mnpq} \left\langle g_j^{(1)} g_l^{(1)} g_n^{(1)} g_q^{(1)} g_j^{(2)} g_l^{(2)} g_n^{(2)} g_q^{(2)} a_{ji}^{(1)} a_{jk}^{(2)} a_{lk}^{(2)} a_{lm}^{(1)} a_{nm}^{(1)} a_{np}^{(2)} a_{qp}^{(2)} a_{qi}^{(1)} \right\rangle$$

$$= \frac{1}{\alpha^4 N_s N_x^4} \sum_{ijkl}\sum_{mnpq} \left\langle g_j^{(1)} g_l^{(1)} g_n^{(1)} g_q^{(1)} g_j^{(2)} g_l^{(2)} g_n^{(2)} g_q^{(2)} \right\rangle$$

$$\times \left( [\delta_{ik}\delta_{km}\delta_{mp} + \delta_{ik}\delta_{lq}\delta_{mp} + \delta_{km}\delta_{jn}\delta_{ip}]\rho_a^4 + [\delta_{ik}\delta_{km}\delta_{nq} + \delta_{km}\delta_{mp}\delta_{jq} + \delta_{mp}\delta_{pi}\delta_{jl} + \delta_{pi}\delta_{ik}\delta_{ln}]\rho_a^2 \right)$$

$$+ \mathcal{O}\left(\frac{1}{\alpha N_x}\right)$$

$$= \tilde{\alpha}^4 \rho_a^4 + \frac{N_s}{\alpha N_x}\tilde{\alpha}^3(4\rho_a^2 + 2\rho_a^4) + \mathcal{O}\left(\frac{1}{\alpha N_x}\right), \tag{38}$$

Here, we applied Isserlis' theorem again, then retained the terms up to the next-to-leading order with respect to $\frac{N_s}{\alpha N_x}$. Similarly, the third term of Eq. 36 becomes

$$\frac{1}{N_s}\left(\frac{N_s}{\alpha N_x}\right)^2 \left\langle \text{tr}\left[A_1^T D_1 D_2 A_2 A_2^T D_2 D_1 A_1\right] \right\rangle = \tilde{\alpha}^2 \rho_a^2 + \frac{\tilde{\alpha} N_s}{\alpha N_x}\left(1 + \frac{\rho_a^2}{N_s}\right). \tag{39}$$

Lastly, the cross-term is evaluated as

$$\frac{1}{N_s}\left(\frac{N_s}{\alpha N_x}\right)^3 \left\langle \text{tr}\left[A_2^T D_2 D_1 A_1 A_1^T D_1 D_2 A_2 A_2^T D_2 D_1 A_1\right] \right\rangle$$

$$= \frac{1}{\alpha^3 N_s N_x^3}\sum_{ijk}\sum_{lmn} \left\langle g_j^{(1)} g_l^{(1)} g_n^{(1)} g_j^{(2)} g_l^{(2)} g_n^{(2)} \right\rangle \left(\delta_{ik}\delta_{km}\rho_a^3 + \delta_{ik}\delta_{ln}\rho_a + \delta_{im}\delta_{jl}\rho_a + \delta_{mk}\delta_{jn}\rho_a^3\right) + \mathcal{O}\left(\frac{1}{\alpha N_x}\right)$$

$$= \tilde{\alpha}^3 \rho_a^3 + \frac{N_s}{\alpha N_x}\tilde{\alpha}^2(2\rho_a + \rho_a^3) + \mathcal{O}\left(\frac{1}{\alpha N_x}\right). \tag{40}$$

Therefore, up to the leading order, the retention performance follows

$$\Delta\epsilon_{RT} = 1 - \tilde{\alpha}^2 \rho_a^2\left(\tilde{\alpha}^2 \rho_a^2 - 2\tilde{\alpha}\rho_a\rho_b + 1\right) + \mathcal{O}\left(\frac{N_s}{\alpha N_x}\right). \tag{41}$$

**Vanilla model** In the vanilla model, we have $\tilde{\alpha} = 1$. Therefore, at $N_s \ll N_x$ limit, from Eqs. 34 and 41, the transfer and retention performance follow

$$\Delta\epsilon_{TF} = \rho_a(2\rho_b - \rho_a), \tag{42a}$$

$$\Delta\epsilon_{RT} = 1 - \rho_a^2(\rho_a^2 - 2\rho_a\rho_b + 1). \tag{42b}$$

**Random activity gating** Under the random task-dependent activity gating, the gating level $\tilde{\alpha} = \alpha$ is a constant. If $N_s \ll \alpha N_x$, the transfer and retention performance are written by

$$\Delta\epsilon_{TF} = \alpha\rho_a(2\rho_b - \alpha\rho_a), \tag{43a}$$

$$\Delta\epsilon_{RT} = 1 - \alpha^2\rho_a^2(\alpha^2\rho_a^2 - 2\alpha\rho_a\rho_b + 1). \tag{43b}$$

Note that, this expression does not hold at the sparse limit $\alpha < \frac{N_s}{N_x}$.

Depending on $(\rho_a, \rho_b)$ combination, the gating level influences the transfer and retention performance in three different manners (Fig. 8). When $\rho_b > \rho_a$, unless both $\rho_a$ and $\rho_b$ are large, there is a monotonic tradeoff between the transfer and retention performance (Fig. 8B). In the red region where $\rho_b \geq \frac{2\rho_a}{3} + \frac{1}{3\rho_a}$ or $\rho_b \geq \frac{2\sqrt{2}}{3} \wedge \rho_a \geq \frac{1}{\sqrt{2}}$, a large gating level above a threshold improves the retention performance (Fig. 8C). When $\rho_b < \rho_a$, high gating level, $\alpha > \frac{\rho_b}{\rho_a}$, lower both transfer and retention performance, thus there is no benefit of choosing large $\alpha$ (Fig. 8D).

**Adaptive activity gating**    In the adaptive task-dependent activity gating model, we introduced a probe trial at the beginning of the task 2 to test the model performance for the new task. Under an adaptive gating with $\rho_g = \max\left\{0, 1 - \frac{\epsilon_{pb}}{\epsilon_o}\right\}$, the probe error $\epsilon_{pb}$ follows:

$$
\begin{aligned}
\langle \epsilon_{pb} \rangle_{A,B,g} &= \left\langle \frac{1}{N_y} \|B_2 - W_1 D_1 A_2\|_F^2 \right\rangle_{A,B,g} \\
&\approx \left\langle \frac{1}{N_y} \left\| B_2 - \frac{N_s}{\alpha N_x} B_1 (D_1 A_1)^T D_1 A_2 \right\|_F^2 \right\rangle_{A,B,g} \\
&= 1 - 2\rho_a \rho_b + \rho_a^2 + \mathcal{O}\left(\frac{N_s}{\alpha N_x}\right).
\end{aligned}
\tag{44}
$$

Thus, at $N_s \ll \alpha N_x$ region, $\rho_g = \max\{0, \rho_a(2\rho_b - \rho_a)\}$, and the effective gating level becomes

$$
\tilde{\alpha} = \begin{cases} \alpha + (1-\alpha)\rho_a (2\rho_b - \rho_a) & (\text{if } 2\rho_b \geq \rho_a) \\ \alpha & (\text{otherwise}) \end{cases}
\tag{45}
$$

meaning that if the transfer performance is positive in the vanilla model, there should be an overlap in the gating, but otherwise, the gating vector for the next task should be sampled independently.

**Average error under a uniform prior on task similarity**    Gating influences the performance differently depending on the feature and readout similarity. Thus, ideally the model should adjust the gating level based on the task similarity, but in most practical settings, the model doesn't know the similarity a priori. Therefore, it is important to analyze how it performs on average under a randomly chosen task similarity. To this end, here we measure the average transfer and retention performance assuming a uniform prior on $0 \leq \rho_a, \rho_b \leq 1$.

In the vanilla model, we have

$$
\Delta \bar{\epsilon}_{TF} = \int_0^1 d\rho_a \int_0^1 d\rho_b \left(\rho_a(2\rho_b - \rho_a)\right) = \frac{1}{6},
\tag{46a}
$$

$$
\Delta \bar{\epsilon}_{RT} = \int_0^1 d\rho_a \int_0^1 d\rho_b \left(1 - \rho_a^2 \left[\rho_a^2 - 2\rho_a \rho_b + 1\right]\right) = \frac{43}{60}.
\tag{46b}
$$

Red and blue dash lines in Figs. 3D, 4C, 4D, 5D, 6C, and 6D represent these baseline performance of the vanilla model: $\Delta \bar{\epsilon}_{TF} = \frac{1}{6}$ and $\Delta \bar{\epsilon}_{RT} = \frac{43}{60}$. By contrast, under a random task-dependent gating,

$$
\Delta \bar{\epsilon}_{TF} = \frac{\alpha}{2} - \frac{\alpha^2}{3}, \quad \Delta \bar{\epsilon}_{RT} = 1 - \frac{\alpha^2}{3} + \frac{\alpha^3}{4} - \frac{\alpha^4}{5}.
\tag{47}
$$

Solid lines in Fig. 3D and dashed lines in Fig. 4B plot the result above. In the case of adaptive gating, the expression of the average performance becomes complicated, yet still numerically tractable.

Note that the average transfer performance is expected to be lower than the retention performance, because forward knowledge transfer requires a high similarity. Even if we use the optimal gating level for transfer, $\alpha^*(\rho_a, \rho_b) = \min(1, \frac{\rho_b}{\rho_a})$, the average transfer performance becomes

$$
\int_0^1 d\rho_a \int_0^1 d\rho_b \alpha^*(\rho_a, \rho_b) \rho_a (2\rho_b - \alpha^*(\rho_a, \rho_b)\rho_a) = \frac{1}{4}.
\tag{48}
$$

## B.2 Plasticity gating

An alternative strategy is to gate plasticity at certain synapses in a task-dependent manner while keeping the activity intact. We implement this method by making synapses from only a subset of input neurons plastic for each task. Given a linear regression model $y = Wx$, the learning dynamics follows:

$$\dot{W}(t) = -\eta(WA_\mu - B_\mu)(D_\mu A_\mu)^T, \tag{49}$$

where $D_\mu = \text{diag}(g_\mu)$ is a diagonal matrix specifying which synapses are gated. As in the activity gating, $g_\mu$ is a gating vector whose elements take either one or zero in a task-dependent manner. We sample elements of $g_\mu$ from a Bernoulli distribution with probability $\alpha$ as before.

Let us denote SVD of $A_\mu$ by $A_\mu = U_\mu \Lambda_\mu V_\mu^T$. Then, from a parallel argument with Eqs. 18 and 19, we have

$$W_\mu = W_{\mu-1} + (B_\mu V_\mu \Lambda_\mu^{-1} - W_{\mu-1} U_\mu)(U_\mu^T D_\mu U_\mu)^{-1} U_\mu^T D_\mu. \tag{50}$$

Notably, unlike Eq. 20b, gating term $D_\mu$ appears both inside and outside of the inverse term ($(U_\mu^T D_u U_\mu)^{-1}$ and $U_\mu^T D_\mu$). Thus, up to the leading order, gating level $\alpha$ does not affect the transfer and retention performance.

## B.3 Input soft-thresholding

Input-dependent gating is an another method proposed for efficient continual learning. In a regression setting, it corresponds to soft-thresholding of the inputs. Let us introduce a soft-thresholding function $\varphi(x)$ by

$$\varphi(x) = sgn(x)\max\{0, |x| - h\}, \tag{51}$$

where $sgn(x)$ represents the sign of $x$. This nonlinearity makes the analysis difficult, but if $N_s \gg 1$ in addition to Assumptions I and II, the error becomes tractable. Note that $N_s \gg 1$ assumption is required only in this subsection. Under the student network defined by

$$\hat{y} = W\varphi(x), \tag{52}$$

the mean-squared error follows

$$\epsilon[W] = \frac{1}{N_y}\left\langle \|Bs\|^2 \right\rangle_s + \frac{1}{N_y}\left\langle \|W\varphi(As)\|^2 \right\rangle_s - \frac{2}{N_y}\left\langle \text{tr}\left[Bs\varphi(As)^T W^T\right]\right\rangle_s. \tag{53}$$

Let us denote $m_i \equiv \sum_{j=1}^{N_s} A_{ij}s_j$, then we have

$$\langle m_i \rangle = 0, \quad \langle m_i^2 \rangle = \sum_{j=1}^{N_s} A_{ij}^2 \approx 1, \quad \langle s_j m_i \rangle = A_{ij}. \tag{54}$$

Thus, $s_j$ and $m_i$ approximately follows a joint Gaussian distribution:

$$\begin{pmatrix} s_j \\ m_i \end{pmatrix} \sim \mathcal{N}\left(\begin{pmatrix} 0 \\ 0 \end{pmatrix}, \begin{pmatrix} 1 & A_{ji} \\ A_{ji} & 1 \end{pmatrix}\right). \tag{55}$$

Denoting $A_{ji} = \rho$ for brevity, we have

$\langle s_j \varphi(m_i) \rangle$

$$= \int_{-\infty}^{\infty} ds \left(\int_{-\infty}^{-h} dm(m+h) + \int_h^{\infty} dm(m-h)\right) \frac{s}{2\pi\sqrt{1-\rho^2}} \exp\left(-\frac{1}{2(1-\rho^2)}\left[s^2 + m^2 - 2\rho sm\right]\right)$$

$$= \int_{-\infty}^{\infty} ds \left(\int_{-\infty}^{-h} dm(m+h) + \int_h^{\infty} dm(m-h)\right) \frac{s}{2\pi}\left[1 + \rho sm + \frac{\rho^2}{2}(s^2-1)(m^2-1) + \mathcal{O}(\rho^3)\right]$$

$$\times \exp\left(-\frac{s^2+m^2}{2}\right)$$

$$= \text{erfc}\left[\frac{h}{\sqrt{2}}\right]\rho + \mathcal{O}(\rho^3). \tag{56}$$

In the third line, we performed a Taylor expansion around $\rho = 0$. Similarly, the expectation of $\varphi(m_j)\varphi(m_i)$ over $s$ is written as

$$\langle \varphi(m_j)\varphi(m_i)\rangle = \frac{1}{2\pi}\left(\int_{-\infty}^{-h} dm(m+h) + \int_{h}^{\infty} dm(m-h)\right)\left(\int_{-\infty}^{-h} dm'(m'+h) + \int_{h}^{\infty} dm'(m'-h)\right)$$

$$\times \left[1 + \rho_c mm' + \frac{\rho_c^2}{2}(m^2-1)(m'^2-1) + \mathcal{O}(\rho_c^3)\right]\exp\left(-\frac{m^2+m'^2}{2}\right)$$

$$= \left(\operatorname{erfc}\left[\frac{h}{\sqrt{2}}\right]\right)^2 \rho_c + \mathcal{O}(\rho_c^3), \tag{57}$$

where $\rho_c = \sum_{k=1}^{N_s} A_{ik}A_{jk}$. Therefore, if $|\rho| \ll 1$ and $|\rho_c| \ll 1$, the error $\epsilon[W]$ is written as

$$\epsilon[W] = \frac{1}{N_y}\|B\|_F^2 + \frac{\alpha^2}{N_y}\operatorname{tr}\left[WAA^TW^T\right] - \frac{2\alpha}{N_y}\operatorname{tr}[BA^TW^T], \tag{58}$$

where the input activity sparseness $\alpha$ follows $\alpha = \operatorname{erfc}\left[\frac{h}{\sqrt{2}}\right]$. Under this loss function, gradient descent from $W = W_o$ converges to

$$W = W_o(I - UU^T) + \frac{1}{\alpha}BA^+, \tag{59}$$

where $A = U\Lambda V^T$ is the singular value decomposition of $A$. Therefore, from $W_o = O$, the weights after task 1 and task 2 become

$$W_1 = \frac{1}{\alpha}\widetilde{W}_1, \quad W_2 = \frac{1}{\alpha}\widetilde{W}_2, \tag{60}$$

where

$$\widetilde{W}_1 \equiv B_1 A_1^+, \quad \widetilde{W}_2 \equiv B_1 A_1^+\left[I - U_2 U_2^T\right] + B_2 A_2^+. \tag{61}$$

This means that for both tasks $(\mu, \nu = 1, 2)$

$$\epsilon_\mu[W_\nu] = \frac{1}{N_y}\|B_\mu\|_F^2 + \frac{1}{N_y}\operatorname{tr}\left[\widetilde{W}_\nu A_\mu A_\mu^T \widetilde{W}_\nu^T\right] - \frac{2}{N_y}\operatorname{tr}[B_\mu A_\mu^T \widetilde{W}_\nu^T], \tag{62}$$

implying that the error is invariant against input sparseness $\alpha$. Thus, in our problem setting, input sparsification via soft-thresholding doesn't influence knowledge transfer or retention. It also implies that from energy efficiency perspective, soft-thresholding is beneficial because it reduces the number of active neurons while preserving its learning proficiency.

## C  Weight regularization

### C.1  Weight regularization in Euclidean metric

Let us next consider a weight regularization approach by introducing L2 regularization with respect to the weight learned in the previous task. The loss function $\ell_\mu$ for the $\mu$-th task is then given by

$$\ell_\mu = \frac{1}{2}\|B_\mu - WA_\mu\|_F^2 + \frac{\lambda}{2}\|W - W_{\mu-1}\|_F^2. \tag{63}$$

When $\lambda > 0$, there exists a unique solution:

$$W = (B_\mu A_\mu^T + \lambda W_{\mu-1})(A_\mu A_\mu^T + \lambda I_x)^{-1}. \tag{64}$$

Thus, from zero-initialization, $W_1$ and $W_2$ become

$$W_1 = B_1 A_1^T \left(A_1 A_1^T + \lambda I_x\right)^{-1} \tag{65a}$$

$$W_2 = \left(B_2 A_2^T + \lambda B_1 A_1^T [A_1 A_1^T + \lambda I_x]^{-1}\right)(A_2 A_2^T + \lambda I_x)^{-1}. \tag{65b}$$

Under $N_x \gg N_s$, the inverse term is approximated by

$$\left(AA^T + \lambda I_x\right)^{-1} = \frac{1}{\lambda}I_x - \frac{1}{\lambda^2}A\left(I_s + \frac{1}{\lambda}A^T A\right)^{-1}A^T$$

$$\approx \frac{1}{\lambda}\left(I_x - \frac{N_s}{N_x + \lambda N_s}AA^T\right). \tag{66}$$

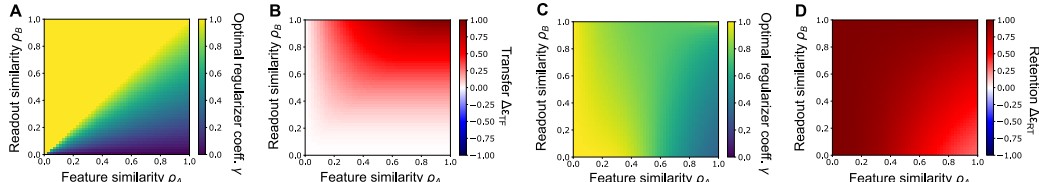

Figure 9: Weight regularization in Euclidean metric. (**A,B**) Optimal regularizer coefficient $\gamma$ that maximizes the transfer performance (**A**), and the maximum performance at the optimal $\gamma$ (**B**) under various ($\rho_a$, $\rho_b$) pairs. (**C,D**) Optimal regularizer coefficient for retention (**C**), and the resultant performance (**D**). Panel C is the same with Fig. 5C (replicated for completeness).

In the first line, we used Woodbury matrix identity, and in the second line, we used $A^T A \approx \frac{N_x}{N_s} I_s$ (see Eq. 99). Let us introduce normalized regularization amplitude $\gamma$ and $\widetilde{\gamma}$ by

$$\gamma \equiv \frac{N_x}{N_x + \lambda N_s}, \quad \widetilde{\gamma} \equiv \frac{N_s}{N_x + \lambda N_s}. \tag{67}$$

Under these approximations, $W_1$ and $W_2$ simplify to

$$W_1 \approx \tfrac{1}{\lambda} B_1 A_1^T \left(1 - \widetilde{\gamma} A_1 A_1^T\right) \approx \widetilde{\gamma} B_1 A_1^T, \tag{68a}$$

$$W_2 \approx \tfrac{1}{\lambda} \left(B_2 A_2^T + \lambda \widetilde{\gamma} B_1 A_1^T\right) \left(I_x - \widetilde{\gamma} A_2 A_2^T\right) \approx \widetilde{\gamma} \left(B_1 A_1^T + B_2 A_2^T\right) - \widetilde{\gamma}^2 B_1 A_1^T A_2 A_2^T. \tag{68b}$$

Here, we further applied $A^T A \approx \frac{N_x}{N_s} I_s$. Note that, unlike Eq. 20b, the result above holds only under Assumptions I & II.

**Transfer performance**   The transfer performance $\Delta\epsilon_{TF}$ follows

$$
\begin{aligned}
\Delta\epsilon_{TF} &= \frac{1}{N_y} \left\langle \|B_2\|_F^2 \right\rangle - \frac{1}{N_y} \left\langle \|B_2 - \widetilde{\gamma} B_1 A_1^T A_2\|_F^2 \right\rangle \\
&= \frac{2\widetilde{\gamma}}{N_y} \left\langle \mathrm{tr}[B_2^T B_1 A_1^T A_2] \right\rangle - \frac{\widetilde{\gamma}^2}{N_y} \left\langle \|B_1 A_1^T A_2\|_F^2 \right\rangle \\
&= \frac{2\widetilde{\gamma}\rho_b}{N_s} \left\langle \mathrm{tr}[A_1^T A_2] \right\rangle - \frac{\widetilde{\gamma}^2}{N_s} \left\langle \|A_1^T A_2\|_F^2 \right\rangle \\
&= \gamma\rho_a \left(2\rho_b - \gamma\rho_a\right) + \mathcal{O}\left(\frac{N_s}{N_x}\right).
\end{aligned}
\tag{69}
$$

In the last line, we used Eq. 33 to estimate $\left\langle \|A_1^T A_2\|_F^2 \right\rangle$. Notably, the expression of the transfer performance, $\Delta\epsilon_{TF}$, coincides with Eq. 5a under $\gamma \to \alpha$, indicating that in terms of transfer performance, the weight regularization with amplitude $\frac{N_x}{N_s}\left(\frac{1}{\alpha} - 1\right)$ is equivalent to random activity gating with gating level $\alpha$. Figs. 9A and B show the optimal regularizer coefficient $\gamma$ that maximizes transfer performance and the resultant performance.

**Retention performance**   The average error on task 1 after learning task 2 is

$$
\begin{aligned}
&\langle \epsilon_1[W_2] \rangle_{A,B} \\
&= \left\langle \frac{1}{N_y} \left\| B_1 - \left(\widetilde{\gamma}[B_1 A_1^T + B_2 A_2^T] - \widetilde{\gamma}^2 B_1 A_1^T A_2 A_2^T\right) A_1 \right\|_F^2 \right\rangle \\
&= \frac{1}{N_y} \left\langle \|B_1(I - \widetilde{\gamma} A_1^T A_1)\|_F^2 \right\rangle + \frac{2\widetilde{\gamma}}{N_y} \left\langle \mathrm{tr}\left[(I - \widetilde{\gamma} A_1^T A_1)B_1^T\left(\widetilde{\gamma} B_1 A_1^T A_2 - B_2\right)A_2^T A_1\right] \right\rangle \\
&\quad + \frac{\widetilde{\gamma}^2}{N_y} \left\langle \|(\widetilde{\gamma} B_1 A_1^T A_2 - B_2)A_2^T A_1\|_F^2 \right\rangle.
\end{aligned}
\tag{70}
$$

Taking the expectation over $A_1, A_2, B_1, B_2$, up to the leading order with respect to $\frac{N_s}{N_x}$, we have

$$\frac{1}{N_y} \left\langle \left\| B_1(I - \widetilde{\gamma} A_1^T A_1) \right\|_F^2 \right\rangle = \frac{1}{N_y} \left\langle \|B_1\|_F^2 - 2\widetilde{\gamma} \mathrm{tr}[B_1^T B_1 A_1^T A_1] + \widetilde{\gamma}^2 \left\| B_1 A_1^T A_1 \right\|_F^2 \right\rangle$$

$$= 1 - \frac{2\widetilde{\gamma}}{N_s} \left\langle \mathrm{tr}[A_1^T A_1] \right\rangle + \frac{\widetilde{\gamma}^2}{N_s} \left\langle \left\| A_1^T A_1 \right\|_F^2 \right\rangle$$

$$= (1 - \gamma)^2 + \mathcal{O}\left(\frac{N_s}{N_x}\right). \tag{71}$$

Similarly, we have

$$\frac{\widetilde{\gamma}}{N_y} \left\langle \mathrm{tr}\left[ (I - \widetilde{\gamma} A_1^T A_1) B_1^T (\widetilde{\gamma} B_1 A_1^T A_2 - B_2) A_2^T A_1 \right] \right\rangle$$

$$= \frac{\widetilde{\gamma}}{N_s} \left\langle \mathrm{tr}\left[ (I - \widetilde{\gamma} A_1^T A_1)(\widetilde{\gamma} A_1^T A_2 - \rho_b I) A_2^T A_1 \right] \right\rangle$$

$$= \rho_a \gamma (1 - \gamma)(\gamma \rho_a - \rho_b) + \mathcal{O}\left(\frac{N_s}{N_x}\right), \tag{72}$$

and

$$\frac{1}{N_y} \widetilde{\gamma}^2 \left\langle \left\| (\widetilde{\gamma} B_1 A_1^T A_2 - B_2) A_2^T A_1 \right\|_F^2 \right\rangle$$

$$= \frac{\widetilde{\gamma}^2}{N_s} \left\langle \mathrm{tr}[A_1^T A_2 A_2^T A_1] - 2\widetilde{\gamma} \rho_b \mathrm{tr}[A_1^T A_2 A_1^T A_2 A_2^T A_1] + \widetilde{\gamma}^2 \mathrm{tr}[A_1^T A_2 A_2^T A_1 A_1^T A_2 A_2^T A_1] \right\rangle$$

$$= \rho_a^2 \gamma^2 \left( 1 - 2\rho_a \rho_b \gamma + \rho_a^2 \gamma^2 \right) + \mathcal{O}\left(\frac{N_s}{N_x}\right). \tag{73}$$

Therefore, the error is written as

$$\langle \epsilon_1[W_2] \rangle = (1 - \gamma)^2 - 2\gamma \rho_a (1 - \gamma)(\rho_b - \gamma \rho_a) + \gamma^2 \rho_a^2 (1 - 2\gamma \rho_a \rho_b + \gamma^2 \rho_a^2). \tag{74}$$

The optimal $\gamma$ for each task similarity $(\rho_a, \rho_b)$ and the resultant retention performance are plotted in Figs. 9C and D.

## C.2 Elastic weight regularization

In the previous section, we regularized the change in the weight matrix in the Euclidean space. However, previous works indicate that the weight should be regularized using the Fisher information matrix (FIM) as the metric [30, 67]. Let us construct an inference model by $\boldsymbol{y} = W\boldsymbol{x} + \sigma \boldsymbol{\xi}$, where $\boldsymbol{\xi}$ is a zero mean Gaussian random variable. Given input-target pair $\boldsymbol{x}, \boldsymbol{y}$, the likelihood of weight $W$ is

$$p(W|\boldsymbol{x}, \boldsymbol{y}) \propto p(\boldsymbol{x}, \boldsymbol{y}|W)p(W) \propto \exp\left( -\frac{1}{2\sigma^2} \|W\boldsymbol{x} - \boldsymbol{y}\|^2 \right). \tag{75}$$

Thus, $(ij, kl)$-th component of FIM becomes

$$\left\langle \frac{\partial^2}{\partial w_{ij} w_{kl}} (-\log p(W|\boldsymbol{x}, \boldsymbol{y})) \right\rangle_{\boldsymbol{s}} = \frac{1}{2\sigma^2} \delta_{ik} \left\langle \sum_m \sum_n a_{jm} a_{ln} s_m s_n \right\rangle_{\boldsymbol{s}} = \frac{1}{2\sigma^2} \delta_{ik} \sum_n a_{jn} a_{ln}, \tag{76}$$

and the quadratic weight regularization in the Fisher information metric follows

$$\sum_{ij} \sum_{kl} \left( \delta_{ik} \sum_n a_{jn} a_{ln} \right) (w_{ij} - w_{ij}^o)(w_{kl} - w_{kl}^o) = \|(W - W_o)A\|_F^2. \tag{77}$$

Therefore, the loss function for this weight regularizer is written as

$$\ell_\mu = \frac{1}{2} \|B_\mu - W A_\mu\|_F^2 + \frac{\lambda}{2} \|(W - W_{\mu-1})A_{\mu-1}\|_F^2. \tag{78}$$

As we will see, this loss function has different forms of solution depending on whether $\rho_a < 1$ or $\rho_a = 1$. We thus consider these two conditions separately below.

### C.2.1 Variable features ($\rho_a < 1$)

Taking gradient with respect to $W$, we have

$$\frac{\partial \ell_\mu}{\partial W} = -(B_\mu - W A_\mu) A_\mu^T + \lambda(W - W_{\mu-1}) A_{\mu-1} A_{\mu-1}^T, \tag{79}$$

indicating that $W$ is written as $W = W_{\mu-1} + Q\widetilde{A}^T$ where $Q$ is a $N_y \times 2N_s$ matrix and $\widetilde{A} \equiv [A_{\mu-1}, A_\mu]$ is an $N_x \times 2N_s$ matrix. Solving $\frac{\partial \ell_\mu}{\partial W} = 0$, we get

$$Q \begin{pmatrix} \lambda A_{\mu-1}^T A_{\mu-1} & A_{\mu-1}^T A_\mu \\ \lambda A_\mu^T A_{\mu-1} & A_\mu^T A_\mu \end{pmatrix} \begin{pmatrix} A_{\mu-1}^T \\ A_\mu^T \end{pmatrix} = \begin{pmatrix} O & B_\mu - W_{\mu-1} A_\mu \end{pmatrix} \begin{pmatrix} A_{\mu-1}^T \\ A_\mu^T \end{pmatrix}, \tag{80}$$

Multiplying both sides with $(A_{\mu-1} \ A_\mu)$ from the right side,

$$Q \begin{pmatrix} \lambda A_{\mu-1}^T A_{\mu-1} & A_{\mu-1}^T A_\mu \\ \lambda A_\mu^T A_{\mu-1} & A_\mu^T A_\mu \end{pmatrix} \begin{pmatrix} A_{\mu-1}^T A_{\mu-1} & A_{\mu-1}^T A_\mu \\ A_\mu^T A_{\mu-1} & A_\mu^T A_\mu \end{pmatrix}$$

$$= \begin{pmatrix} O & B_\mu - W_{\mu-1} A_\mu \end{pmatrix} \begin{pmatrix} A_{\mu-1}^T A_{\mu-1} & A_{\mu-1}^T A_\mu \\ A_\mu^T A_{\mu-1} & A_\mu^T A_\mu \end{pmatrix} \tag{81}$$

If $\lambda \neq 0$ and $\rho_a < 1$, two square matrices in the above equations are almost surely invertible under assumptions I & II. Therefore, we have

$$Q = \begin{pmatrix} O & B_\mu - W_{\mu-1} A_\mu \end{pmatrix} \begin{pmatrix} \lambda A_{\mu-1}^T A_{\mu-1} & A_{\mu-1}^T A_\mu \\ \lambda A_\mu^T A_{\mu-1} & A_\mu^T A_\mu \end{pmatrix}^{-1} \tag{82}$$

Under $N_x \gg N_s$, from Eq. 99, the inverse matrix is approximated by

$$\begin{pmatrix} \lambda A_{\mu-1}^T A_{\mu-1} & A_{\mu-1}^T A_\mu \\ \lambda A_\mu^T A_{\mu-1} & A_\mu^T A_\mu \end{pmatrix}^{-1} \approx \frac{N_s}{N_x} \begin{pmatrix} \lambda I_s & \rho_a I_s \\ \lambda \rho_a I_s & I_s \end{pmatrix}^{-1} = \frac{N_s}{N_x(1-\rho_a^2)} \begin{pmatrix} \frac{1}{\lambda} I_s & -\frac{\rho_a}{\lambda} I_s \\ -\rho_a I_s & I_s \end{pmatrix} \tag{83}$$

Therefore, $W = W_{\mu-1} + Q\widetilde{A}^T$ follows

$$W = W_{\mu-1} \left( I - \frac{N_s}{N_x(1-\rho_a^2)} A_\mu \left[ A_\mu^T - \rho_a A_{\mu-1}^T \right] \right) + \frac{N_s}{N_x(1-\rho_a^2)} B_\mu \left( A_\mu^T - \rho_a A_{\mu-1}^T \right). \tag{84}$$

Notably, the equation above doesn't depend on the regularizer amplitude $\lambda$, except for $\lambda \neq 0$ condition. At $\lambda \to 0$ limit, the inverse matrix term in Eq. 82 effectively becomes singular, and thus the equation above no longer holds.

Let us suppose the first task is learned without any weight regularization, or learned with weight regularization in the Fisher information metric imposed by an uncorrelated task. Then, we have $W_1 = \frac{N_s}{N_x} B_1 A_1^T$. Hence, the transfer performance becomes the same with the vanilla model. The weight after the second task becomes

$$W_2 = \frac{N_s}{N_x} B_1 A_1^T \left( I - \frac{N_s}{N_x(1-\rho_a^2)} A_2 [A_2^T - \rho_a A_1^T] \right) + \frac{N_s}{N_x(1-\rho_a^2)} B_2 (A_2^T - \rho_a A_1^T). \tag{85}$$

Thus, the retention performance under $W_2$ follows

$$\Delta \epsilon_{RT} = \frac{1}{N_y} \left\langle \|B_1\|_F^2 \right\rangle$$
$$- \frac{1}{N_y} \left\langle \left\| B_1 \left( I - \frac{N_s}{N_x} A_1^T A_1 \right) + \frac{N_s}{N_x(1-\rho_a^2)} \left( \frac{N_s}{N_x} B_1 A_1^T A_2 - B_2 \right) (A_2^T - \rho_a A_1^T) A_1 \right\|_F^2 \right\rangle. \tag{86}$$

Taking expectation over $B_1$ and $B_2$, $\Delta \epsilon_{RT}$ is rewritten as

$$\Delta \epsilon_{RT} = 1 - \frac{1}{N_s} \left\langle \left\| I - \frac{N_s}{N_x} A_1^T A_1 \right\|_F^2 \right\rangle$$
$$- \frac{2}{N_x(1-\rho_a^2)} \left\langle \operatorname{tr} \left[ \left( I - \frac{N_s}{N_x} A_1^T A_1^T \right) \left( \frac{N_s}{N_x} A_1^T A_2 - \rho_b I \right) (A_2^T - \rho_a A_1^T) A_1 \right] \right\rangle$$
$$- \frac{1}{N_s} \left( \frac{N_s}{N_x(1-\rho_a^2)} \right)^2 \left\langle \left\| \frac{N_s}{N_x} A_1^T A_2 (A_2^T - \rho_a A_1^T) A_1 \right\|_F^2 + \left\| (A_2^T - \rho_a A_1^T) A_1 \right\|_F^2 \right\rangle$$
$$+ \frac{2\rho_b}{N_x} \left( \frac{N_s}{N_x(1-\rho_a^2)} \right)^2 \left\langle \operatorname{tr} \left[ A_1^T (A_2 - \rho_a A_1) A_2^T A_1 (A_2^T - \rho_a A_1^T) A_1 \right] \right\rangle. \tag{87}$$

The second term is rewritten as

$$\frac{1}{N_s}\left\langle \left\| I - \frac{N_s}{N_x}A_1^T A_1 \right\|_F^2 \right\rangle = 1 - \frac{2}{N_x}\left\langle \text{tr}[A_1^T A_1] \right\rangle + \frac{1}{N_s}\left(\frac{N_s}{N_x}\right)^2 \left\langle \text{tr}[A_1^T A_1 A_1^T A_1] \right\rangle = \frac{N_s+1}{N_x}. \quad (88)$$

Moreover, $(A_2^T - \rho_a A_1^T)A_1$ term also cancels out up to the leading order term because

$$\frac{1}{N_s}\left(\frac{N_s}{N_x(1-\rho_a^2)}\right)^2 \left\langle \left\|(A_2^T - \rho_a A_1^T)A_1\right\|_F^2 \right\rangle$$

$$= \frac{1}{N_s}\left(\frac{N_s}{N_x(1-\rho_a^2)}\right)^2 \sum_{ijkl} \left\langle a_{ji}^{(1)}(a_{jk}^{(2)} - \rho_a a_{jk}^{(1)})(a_{lk}^{(2)} - \rho_a a_{lk}^{(1)})a_{li}^{(1)} \right\rangle$$

$$= \frac{1}{N_s}\left(\frac{N_s}{N_x(1-\rho_a^2)}\right)^2 \sum_{ijkl} \left\langle a_{ji}^{(1)} a_{li}^{(1)} \right\rangle \left\langle (a_{jk}^{(2)} - \rho_a a_{jk}^{(1)})(a_{lk}^{(2)} - \rho_a a_{lk}^{(1)}) \right\rangle$$

$$= \frac{1}{N_s}\left(\frac{N_s}{N_x(1-\rho_a^2)}\right)^2 \sum_{ijkl} \delta_{jl} \frac{1-\rho_a^2}{N_s^2} = \frac{N_s}{N_x(1-\rho_a^2)} \quad (89)$$

Similarly, the trace terms are also cancelled out up to the leading order term:

$$\frac{1}{N_x}\left(\frac{N_s}{N_x(1-\rho_a^2)}\right)^2 \left\langle \text{tr}[A_1^T(A_2^T - \rho_a A_1)A_2^T A_1(A_2^T - \rho_a A_1^T)A_1] \right\rangle$$

$$= \frac{1}{N_x}\left(\frac{N_s}{N_x(1-\rho_a^2)}\right)^2 \sum_{ijklmn} \left\langle a_{ji}^{(1)}(a_{jk}^{(2)} - \rho_a a_{jk}^{(1)})a_{lk}^{(2)} a_{lm}^{(1)}(a_{nm}^{(2)} - \rho_a a_{nm}^{(1)})a_{ni}^{(1)} \right\rangle$$

$$= \frac{1}{N_x}\left(\frac{N_s}{N_x(1-\rho_a^2)}\right)^2 \sum_{ijklmn} \left\langle (a_{jk}^{(2)} - \rho_a a_{jk}^{(1)})(a_{nm}^{(2)} - \rho_a a_{nm}^{(1)}) \right\rangle \left\langle a_{ji}^{(1)} a_{lk}^{(2)} a_{lm}^{(1)} a_{ni}^{(1)} \right\rangle$$

$$= \frac{1}{N_x}\left(\frac{N_s}{N_x(1-\rho_a^2)}\right)^2 \sum_{ijklmn} \frac{1}{N_s}\delta_{jn}\delta_{km}(1-\rho_a^2)\frac{1}{N_s^2}\left(\rho_a + \delta_{jl}\delta_{ik}2\rho_a\right)$$

$$= \frac{N_s\rho_a}{N_x(1-\rho_a^2)}\left(1 + \frac{2}{N_x N_s}\right), \quad (90)$$

and

$$\frac{1}{N_x(1-\rho_a^2)}\left\langle \text{tr}[(I - \frac{N_s}{N_x}A_1^T A_1)(\frac{N_s}{N_x}A_1^T A_2 - \rho_b I)(A_2^T - \rho_a A_1^T)A_2] \right\rangle$$

$$= \frac{1}{N_x(1-\rho_a^2)}\sum_{ijkl} \left\langle \left(\delta_{ij} - \frac{N_s}{N_x}\sum_m a_{mi}^{(1)} a_{mj}^{(1)}\right)\left(\frac{N_s}{N_x}\sum_n a_{nj}^{(1)} a_{nk}^{(2)} - \rho_b \delta_{jk}\right)\left(a_{lk}^{(2)} - \rho_a a_{lk}^{(1)}\right) a_{li}^{(2)} \right\rangle = 0. \quad (91)$$

Thus, we get $\Delta\epsilon_{RT} = 1 - \mathcal{O}\left(\frac{N_s}{N_x(1-\rho_a^2)}\right)$. Therefore, if $N_s \ll (1-\rho_a^2)N_x$, we have $\Delta\epsilon_{RT} \approx 1$ regardless of task similarity $\rho_a$ and $\rho_b$.

### C.2.2 Fixed features ($\rho_a = 1$)

When $\rho_a = 1$, $A_{\mu-1} = A_\mu = A$. The gradient follows

$$\frac{\partial \ell_\mu}{\partial W} = [-(B_\mu - WA) + \lambda(W - W_{\mu-1})A]\, A^T, \quad (92)$$

and the weight $W$ is written as $W = W_{\mu-1} + QU^T$, where $U$ is defined by SVD of A: $A = U\Lambda V^T$. Solving $\frac{\partial \ell_\mu}{\partial W} = 0$, we get

$$W = W_{\mu-1}\left(I - \frac{1}{1+\lambda}UU^T\right) + \frac{1}{1+\lambda}B_\mu A^+ \quad (93)$$

Thus, the weight after task 1 and 2 follow

$$W_1 = \frac{1}{1+\lambda}B_1 A^+, \quad W_2 = \frac{1}{1+\lambda}B_1 A^+\left(I - \frac{1}{1+\lambda}UU^T\right) + \frac{1}{1+\lambda}B_2 A^+. \quad (94)$$

Therefore, the transfer performance is

$$\Delta \epsilon_{TF} = \frac{1}{N_y} \left\langle \|B_2\|^2 \right\rangle - \frac{1}{N_y} \left\langle \|B_2 - W_1 A_2\|^2 \right\rangle$$
$$= \frac{2\rho_b}{1+\lambda} - \frac{1}{(1+\lambda)^2} = \gamma_f \left(2\rho_b - \gamma_f\right). \tag{95}$$

In the last line, we defined $\gamma_f$ by $\gamma_f \equiv \frac{1}{1+\lambda}$. Similarly, the retention performance is written as

$$\Delta \epsilon_{RT} = \frac{1}{N_y} \left\langle \|B_1\|_F^2 \right\rangle - \frac{1}{N_y} \left\langle \|B_1 - W_2 A\|_F^2 \right\rangle$$
$$= \frac{1}{N_y} \left\langle \|B_1\|_F^2 \right\rangle - \frac{1}{N_y} \left\langle \left\| \left(1 - \frac{\lambda}{(1+\lambda)^2}\right) B_1 - \frac{1}{1+\lambda} B_2 \right\|_F^2 \right\rangle$$
$$= 2\gamma_f (1 - \gamma_f)^2 - \gamma_f^4 + 2\gamma_f \left[(1-\gamma_f)^2 - \gamma_f^2\right] \rho_b. \tag{96}$$

If the first task is learned without weight regularization, the retention performance instead becomes:

$$\Delta \epsilon_{RT} = \frac{1}{N_y} \left\langle \|B_1\|_F^2 \right\rangle - \frac{1}{N_y} \left\langle \left\| \frac{1}{1+\lambda} (B_1 - B_2) \right\|_F^2 \right\rangle = 1 - 2\gamma_f^2 (1 - \rho_b). \tag{97}$$

## D  Properties of very tall zero-mean random matrices

Our simple analytical results rely on the assumption that matrix $A \in \mathbb{R}^{N_x \times N_s}$ is very tall (i.e., $N_x \gg N_s$), and each element of $A$ is sampled independently from a normal distribution with mean zero and a finite variance.

Given $A_{ij} \sim \mathcal{N}(0, \frac{1}{N_s})$, we have $\left\langle \frac{1}{N_x} A^T A \right\rangle_A = \frac{1}{N_s} I_s$. The element-wise deviation from the mean is evaluated as

$$\left\langle \left( \left[ \frac{1}{N_x} A^T A - \frac{1}{N_s} I_s \right]_{ij} \right)^2 \right\rangle_A = \frac{1 + \delta_{ij}}{N_x N_s^2}, \tag{98}$$

implying that $\left\langle \left\| \frac{1}{N_x} A^T A - \frac{1}{N_s} I_s \right\|_F^2 \right\rangle_A = \frac{1}{N_x} \left(1 + \frac{1}{N_s}\right)$. Therefore, at $N_x \gg N_s$ limit, $\frac{1}{N_x} A^T A$ approximately follows

$$\frac{1}{N_x} A^T A \approx \frac{1}{N_s} I_s. \tag{99}$$

Even in the presence of random gating, if the effective matrix is still tall (i.e., $\alpha N_x \gg N_s$), similar approximations hold. Given a diagonal matrix $D$ whose diagonal components are sampled independently from a Bernoulli distribution with rate $\alpha$, at $\alpha N_x \gg N_s$ limit, we have

$$\frac{1}{\alpha N_x} (DA)^T DA \approx \frac{1}{N_s} I_s. \tag{100}$$

This result implies that,

$$(DA)^+ \approx \frac{N_s}{\alpha N_x} (DA)^T. \tag{101}$$

Moreover, denoting the SVD of $DA$ by $DA = U\Sigma V^T$, at $\alpha N_x \gg N_s$ limit, we have

$$UU^T \approx \frac{N_s}{\alpha N_x} DAA^T D. \tag{102}$$

This is because, at $\alpha N_x \gg N_s$ limit, from Eq. 100, the eigenspectrum of matrix $\frac{1}{\alpha N_x} DA(DA)^T$ is concentrated at $\lambda = \frac{1}{N_s}$, and thus,

$$\frac{N_s}{\alpha N_x} DAA^T D = \frac{N_s}{\alpha N_x} U\Sigma^2 U^T \approx UU^T. \tag{103}$$

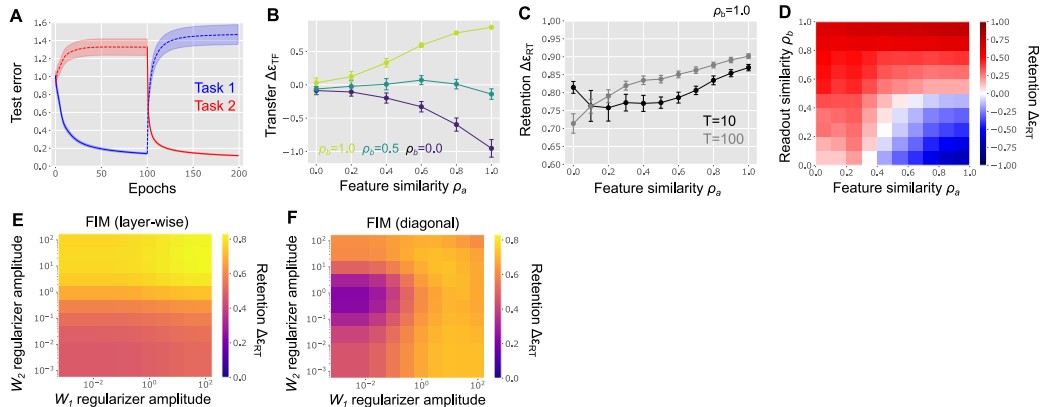

Figure 10: Permuted MNIST with a latent variable. (**A**) Learning curves under $(\rho_a, \rho_b) = (0.8, 0.2)$. (**B**) Knowledge transfer under various task similarities. (**C**) Comparison of the retention performance measured after 10 epochs (black) and 100 epochs (gray) of training. (**D**) Retention performance after 100 epochs of training with task 2. After 100 epochs of training, the system still exhibits strong asymmetric dependence on the feature and readout similarities, but the non-monotonic dependence on the feature similarity under $\rho_b \approx 1$ region disappears. (**E,F**) Retention performance under the weight regularization in the Fisher information metric (FIM) with a layer-wise approximation (panel E) and a diagonal approximation (panel F), under $(\rho_a, \rho_b) = (0.5, 0.5)$. Unlike Figs. 7F-H in the main figure, we adjusted the regularizer amplitude of two layers independently. As in Fig. 7, error bars in panel A-C represent standard errors over 10 random seeds.

# E    Numerical methods

Numerical experiments were conducted in standard laboratory GPUs and CPUs. Source codes for all numerical results are made publicly available at `https://github.com/nhiratani/transfer_retention_model`.

## E.1    Implementation of linear teacher-student models

Numerical results in Figs. 2-6 and 8-9 were implemented as below. We set the latent variable dimensionality $N_s = 30$, the input width $N_x = 3000$, and the output width $N_y = 10$. The student weight $W$ was initialized as the zero matrix, and updated with the full gradient descent with learning rate $\eta = 0.001$. We trained the network with the first task for $N_{epoch}$ epochs, and then retrained it with the second task for another $N_{epoch}$ epochs. We used $N_{epoch} = 100$ for the vanilla model and the activity gating models, but we used $N_{epoch} = 500$ for other models. Unless stated otherwise, error bars represent standard deviation over 10 random seeds. Average performances in Figs. 3D, 4B-D, 5D, and 6CD were estimated by taking average over 100 pairs of $(\rho_a, \rho_b)$, sampled uniformly from $\rho_a, \rho_b \in [0, 1]$.

In the input soft-thresholding model, we estimated the gradient in a sample-based manner because the exact gradient is not tractable due to nonlinearity. We estimated the gradient from 10000 random samples at each iterations, then updated the model for 5000 iterations with learning $\eta = 0.01$.

## E.2    Implementation of permuted MNIST with latent

**Data generation**    We used permuted MNIST dataset [34, 22], a common benchmark for continual learning, but with addition of the latent space. We constructed a four dimensional latent space $s$ using the binary representation of digits, $s_0 = [0, 0, 0, 0]^T$, $s_1 = [0, 0, 0, 1]^T$, and so on. The target output was generated by a projection of the latent variable $s$ to a ten-dimensional space, $y^* = B(s - \frac{1}{2}\mathbf{1})$, where $B$ is a $10 \times 4$ matrix generated randomly. The elements of matrix $B_1$ for the first task were sampled independently from a Gaussian distribution with mean zero and variance one. For the second task, we introduced readout similarity by keeping some of the elements while resampling other elements. We defined the readout similarity $\rho_b$ by the fraction of the elements kept the same between two tasks.

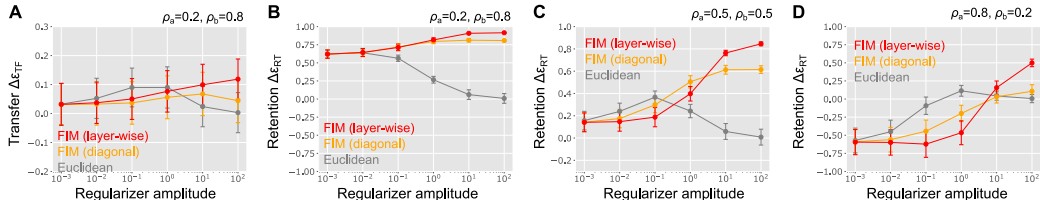

Figure 11: Effect of layer-wise weight regularization in deep feedforward networks solving permuted MNIST. **(A)** The transfer performance under three forms of weight regularization as a function of the regularization amplitude. **(B-D)** The same as A, but the retention performance were plotted. The panels are parallel to Fig. 7E-H, but here, we used a network with 3 hidden layers instead of a one-hidden layer network used for Fig. 7. When the regularizer amplitude is larger than $10^2$, we observed instability in learning dynamics.

Feature similarity was introduced by permuting the input pixels as done previously [22]. We used the vanilla MNIST images for the first task and permuted some of the pixels in the second task depending on the feature similarity $\rho_a$.

**Model implementation**   We used one hidden layer neural network with ReLU non-linearity in the hidden layer: $\boldsymbol{y} = \boldsymbol{b}_2 + W_2 \text{ReLU} [\boldsymbol{b}_1 + W_1 \boldsymbol{x}]$. We set the hidden layer width to $N_h = 1500$. The input and output widths were set to $N_x = 784$ and $N_y = 10$. We initialized the first weight $W_1$ and the bias $\boldsymbol{b}_1$ by sampling each weight independently from a Gaussian distribution with mean zero and variance $\frac{1}{N_h^2}$. We initialized the second weight $W_2$ and the bias $\boldsymbol{b}_2$ in the same manner, but with variance $\frac{1}{N_y^2}$. Here, we set the amplitude of the initial weights small to operate the learning in the rich regime [53]. We set the mini-batch size to 300 and the learning rate to $\eta = 0.01$, and trained the network for 100 epochs per task. The retention performance was measured after 10 epochs of training with task 2 except for Figs. 10 C, 10 D and 11 A-D where the retention was evaluated after 100 epochs. In the learning of task 2, the test error typically drops significantly in the first 10 epochs and shifts to gradual improvement afterward (red line in Fig. 10A). Notably, the asymmetric task similarity dependence of the retention performance was observed consistently both after a short and long training on the second task (Figs. 7B and 10D).

**Weight regularization in the Fisher information metric**   From the same argument made in section C.2, FIM of a noise-free model is approximated by the Hessian of the mean squared error. Let us consider a vanilla feedforward neural network with depth $K$, where hidden layer activity $\boldsymbol{h}_1, ..., \boldsymbol{h}_{K-1}$ and the output $\boldsymbol{y}$ follow

$$\boldsymbol{h}_k = \phi\left(\boldsymbol{b}_k + W_k \boldsymbol{h}_{k-1}\right), \text{ for } k = 1, ..., K-1 \tag{104}$$

$$\boldsymbol{y} = \boldsymbol{b}_K + W_K \boldsymbol{h}_{k-1}, \tag{105}$$

where $\phi$ is an element-wise rectified linear function. For brevity, we denote $\nabla\phi_\mu$ as a diagonal matrix where the diagonal components represent element-wise derivative $\phi'(\boldsymbol{b}_k + W_k \boldsymbol{h}_{k-1})$. Given loss function $\ell = \frac{1}{2}\|\boldsymbol{y} - \boldsymbol{y}^*\|^2$, the Hessian is estimated as

$$
\begin{aligned}
\frac{\partial^2 \ell}{\partial w_{ij}^{(\mu)} \partial w_{kl}^{(\mu)}} &= \frac{\partial \ell}{\partial w_{ij}^{(\mu)}} \left( \left[\nabla\phi_\nu W_{\nu+1}^T ... \nabla\phi_{K-1} W_K^T (\boldsymbol{y} - \boldsymbol{y}^*)\right]_k h_{\nu-1,l}\right) \\
&\approx \left[\nabla\phi_\nu W_{\nu+1}^T ... \nabla\phi_{K-1} W_K^T W_K \nabla\phi_{K-1} ... W_{\mu+1} \nabla\phi_\mu\right]_{ki} h_{\mu-1,j} h_{\nu-1,l} \\
&\quad + [\mu > \nu]_+ \left[\nabla\phi_\nu W_{\nu+1}^T ... \nabla\phi_{\mu-1}\right]_{kj} \left[\nabla\phi_\mu W_{\mu+1}^T ... W_K^T (\boldsymbol{y} - \boldsymbol{y}^*)\right]_i h_{\nu-1,l} \\
&\quad + [\mu < \nu]_+ \left[\nabla\phi_\mu W_{\mu+1}^T ... \nabla\phi_{\nu-1}\right]_{il} \left[\nabla\phi_\nu W_{\nu+1}^T ... W_K^T (\boldsymbol{y} - \boldsymbol{y}^*)\right]_k h_{\mu-1,j} \\
&\equiv \widetilde{H}_{ijkl}^{(\mu,\nu)}. \tag{106}
\end{aligned}
$$

In the second line, we omitted the second-order derivative $\phi''$ as this term is effectively negligible under ReLU activation function.

Thus, distance between two weights $\delta W = W - W_o$ in the metric defined by this approximated Hessian is written as

$$\sum_{\mu,\nu}\sum_{ijkl}\delta w_{ij}^{(\mu)}\delta w_{kl}^{(\nu)}\widetilde{H}_{ijkl}^{(\mu,\nu)}$$

$$= \sum_{\mu,\nu}\boldsymbol{h}_{\mu-1}^T\delta W_\mu^T\nabla\phi_\mu W_{\mu+1}^T...\nabla\phi_{K-1}W_K^T W_K\nabla\phi_{K-1}...W_{\nu+1}\nabla\phi_\nu\delta W_\nu\boldsymbol{h}_{\nu-1}$$

$$+ 2\sum_{\mu>\nu}\boldsymbol{h}_{\nu-1}^T\delta W_\nu^T\nabla\phi_\nu W_{\nu+1}^T...\nabla\phi_{\mu-1}\delta W_\mu^T\nabla\phi_{\mu+1}W_{\mu+1}^T...W_L^T(\boldsymbol{y}-\boldsymbol{y}^*). \tag{107}$$

We define a layer-wise approximation of the weight regularization in the Fisher information metric by taking $\mu = \nu$ components of the distance defined above:

$$R^{lw}[W] = \frac{1}{2}\sum_{\mu=1}^K\left\langle\left\|W_K\nabla\phi_{K-1}...W_{\mu+1}\nabla\phi_\mu\delta W_\mu\phi_{\mu-1}\right\|^2\right\rangle. \tag{108}$$

Here, the expectation is taken over training data $(\boldsymbol{x}, \boldsymbol{y}^*)$ of the previous task (i.e., the task the network shouldn't forget). Taking the derivative with respect to $\delta W_k$ for $k = 1, .., K$, we have

$$\frac{\partial R^{lw}}{\partial\delta W_\mu} = \left\langle\nabla\phi_\mu W_{\mu+1}^T...\nabla\phi_{K-1}W_K^T W_K\nabla\phi_{K-1}...W_{\mu+1}\nabla\phi_\mu\delta W_\mu\boldsymbol{h}_{\mu-1}\boldsymbol{h}_{\mu-1}^T\right\rangle$$

$$\approx \left\langle\nabla\phi_\mu W_{\mu+1}^T...\nabla\phi_{K-1}W_K^T W_K\nabla\phi_{K-1}...W_{\mu+1}\nabla\phi_\mu\right\rangle\delta W_\mu\left\langle\boldsymbol{h}_{\mu-1}\boldsymbol{h}_{\mu-1}^T\right\rangle. \tag{109}$$

This layer-wise approximation captures the true Fisher information metric more accurately than element-wise approximation implemented previously, while computationally less heavy than the estimation of the true metric. In the numerical estimations, we scaled the regularization terms at the k-the layer by

$$Z_k = \left\|\left\langle\nabla\phi_\mu W_{\mu+1}^T...\nabla\phi_{K-1}W_K^T W_K\nabla\phi_{K-1}...W_{\mu+1}\nabla\phi_\mu\right\rangle\right\|_2\left\|\left\langle\boldsymbol{h}_{\mu-1}\boldsymbol{h}_{\mu-1}^T\right\rangle\right\|_2, \tag{110}$$

where $\|\cdot\|_2$ is the spectral norm. We did not impose weight regularization on the bias parameters $\boldsymbol{b}_k$.

In the case of synapse-wise approximation (i.e., elastic regularization), the regularizer is instead written as

$$R^{sw}[W] = \frac{1}{2}\sum_\mu\sum_{ij}\frac{\partial^2\ell}{\partial(w_{ij}^{(\mu)})^2}\left(\delta w_{ij}^\mu\right)^2$$

$$= \frac{1}{2}\left\langle\left[\nabla\phi_\mu W_{\mu+1}^T...\nabla\phi_{K-1}W_K^T W_K\nabla\phi_{K-1}...W_{\mu+1}\nabla\phi_\mu\right]_{ii}h_{\mu-1,j}^2\right\rangle\left(\delta w_{ij}^{(\mu)}\right)^2. \tag{111}$$

Thus, the gradient of $R^{sw}$ with respect to $\delta W_\mu$ follows

$$\frac{\partial R^{sw}}{\partial(\delta W_\mu)} = \left\langle\text{diag}\left[\nabla\phi_\mu W_{\mu+1}^T...\nabla\phi_{K-1}W_K^T W_K\nabla\phi_{K-1}...W_{\mu+1}\nabla\phi_\mu\right](\boldsymbol{h}_{\mu-1}\odot\boldsymbol{h}_{\mu-1})^T\right\rangle\odot\delta W_\mu. \tag{112}$$

We scaled the regularization term by the largest coefficient in a layer-wise manner.

In the case of one-hidden layered networks, the layer-wise regularizer is written as

$$R^{lw}[W_1, W_2] = \frac{1}{2Z_1}\left\langle\|W_2\nabla\phi\delta W_1\boldsymbol{x}\|^2\right\rangle + \frac{1}{2Z_2}\left\langle\|\delta W_2\phi\|^2\right\rangle, \tag{113}$$

where the normalization constants follow $Z_1 = \|W_2 W_2^T\|_2\|\langle\boldsymbol{x}\boldsymbol{x}^T\rangle\|_2$, $Z_2 = \|\langle\phi\phi^T\rangle\|_2$. By contrast, the synapse-wise regularizer is given as

$$R^{sw}[W_1, W_2] = \frac{1}{2Z_1}\sum_{ij}\sum_k\left(w_{ki}^{(2)}\phi_i'x_j\right)^2\left(\delta w_{ij}^{(1)}\right)^2 + \frac{1}{2Z_2}\sum_{ij}\phi_j^2\left(\delta w_{ij}^{(2)}\right)^2, \tag{114}$$

where $Z_1 = \max_{ij}\left\langle\sum_k(w_{ki}^{(2)}\phi_i'x_j)^2\right\rangle$ and $Z_2 = \max_j\langle\phi_j^2\rangle$. To see the potential effect of different normalization methods used for the layer-wise and diagonal approximations, in Figs. 10 E and F, we

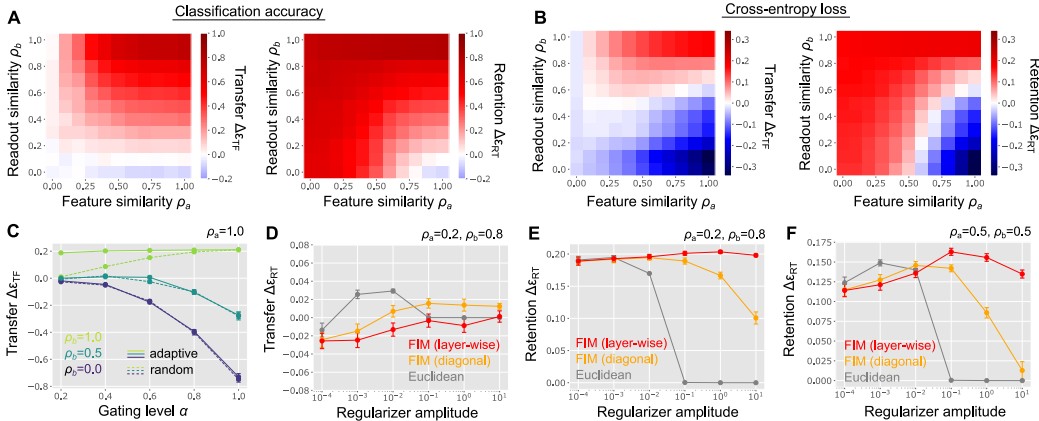

Figure 12: Permuted MNIST with pixel and label permutations. **(A,B)** Transfer and classification performance measured by the classification accuracy (A) and the cross-entropy loss (B). **(C)** Transfer performance of random (dashed lines) and adaptive (solid lines) activity gating models. **(D-F)** Performance of the weight regularization in the Euclidean metric, and approximated Fisher information metrics.

independently changed the regularizer amplitude for $W_1$ and $W_2$ and measured the retention performance under the two approximation methods. Even in this setting, the layer-wise approximation of the Fisher-information metric robustly outperformed the diagonal approximation, when the regulazier amplitude for $W_2$ was sufficiently large.

In Fig. 11, we implemented this method to a feedforward network with three all-to-all hidden layers, where hidden layer widths are set to be 784-1000-300-100-10. We used the learning rate $\eta = 0.01$, mini-batch size 300, and trained the network for 100 epochs per task. The weight regularization based on a layer-wise approximation of the Fisher-information metric, robustly outperformed both the diagonal approximation of the metric and weight regularization in the Euclidean space even in this deep network (Fig. 11B-D).

### E.3    Implementation of permuted MNIST

In Fig. 12, we implemented permuted MNIST without any explicit latent structure. We modulated the input feature similarity by permuting a subset of input pixels between the first and the second tasks, and modulated the readout similarity by permuting a subset of the output labels as shown in Figs. 1C and 1D. We used the same one-hidden layer network with the model above, but introduced the softmax at the output units and estimated the loss using cross-entropy loss. We set the mini-batch size to be 300, and the epoch per task as 100. The retention performance was evaluated at the end of the training of the second task using Eq. 2.

When transfer and retention performance were assessed based on differences in classification accuracy, both measures exhibited a positive monotonic relationship with feature and readout similarity (Fig. 12A). However, when evaluated using cross-entropy loss, the training loss function, transfer and retention performance showed a non-monotonic dependence on feature and readout similarities (Fig. 12B). Specifically, under low readout similarity, increasing feature similarity negatively impacted transfer performance.

Adaptive gating with a probe trial improved transfer performance (Fig. 12C vs. Figs. 7C and 4A). Additionally, weight regularization in the layer-wise approximation of the Fisher information metric yielded better retention performance compared to its diagonal approximation and weight regularization in Euclidean space. (Figs. 12D-F).

