# OpenReview forum: "Disentangling and mitigating the impact of task similarity for continual learning"
_NeurIPS.cc/2024/Conference — NeurIPS 2024 poster_

### Official Review · Reviewer_LdFf · 2024-07-10

**Soundness:** 3
**Presentation:** 3
**Contribution:** 3
**Rating:** 6
**Confidence:** 3

**Summary:**

This paper analyzes the impact of task similarity on Continual Learning (CL) within a linear teacher-student model that incorporates low-dimensional latent structures. The findings indicate that high input feature similarity combined with low readout similarity is detrimental to both knowledge transfer and retention

**Strengths:**

+ The paper is well-motivated, addressing the important problem of understanding the impact of task similarity on continual learning.
+ The authors provide a thorough analysis of the impact of task similarity on CL using a linear teacher-student model with low-dimensional latent structures. The results demonstrate that high feature similarity and low readout similarity are catastrophic for both knowledge transfer and retention.
+ The paper tests its predictions numerically using the permuted MNIST task, further validating the theoretical findings.
+ The authors provide the code, which enhances the reproducibility of their results.

**Weaknesses:**

+ The current theory is constrained to task incremental CL involving two regression tasks. It would be beneficial to explore whether the theory can be generalized to more tasks or other settings, such as classification tasks.
+ How can readout similarity be understood in other CL tasks, such as classification tasks?

**Questions:**

Please refer to the weaknesses.

**Limitations:**

The authors have included a limiation discussion in Sec. 8.

---

> ### Author Rebuttal · Authors · 2024-08-06
>
> Thank you for your insightful comments.
>
> We have included an additional figure to clarify the concept of readout similarity in a classification setting and to demonstrate the applicability of our framework to such settings. The lower half of panel A illustrates two tasks with low readout similarity. In the first task, we use the standard MNIST dataset, where digit images are classified into one of ten labels. For the second task (of a two-task continual learning), we permute half of the labels, resulting in a low readout similarity between the two tasks ($\rho_b=0.5$). Although the partial label permutation introduced here is somewhat artificial, a partial shift in target labels is expectedly a common scenario in real-world continual learning tasks in dynamic environments. Therefore, we believe it is crucial to understand how feature and readout similarity influence continual learning performance.
>
> In panels B and C, we evaluated the knowledge transfer and retention performance under this continual classification task with partial input and label permutations. For this evaluation, we used the one-hidden layer model depicted in Fig. 6 of the manuscript, with modifications: we added a softmax function in the last layer and employed cross-entropy loss for gradient calculation instead of mean-squared error.
>
> Panel B illustrates the knowledge transfer and retention between the two tasks based on changes in classification accuracy, while panel C shows the transfer and retention performance based on changes in cross-entropy loss between the network output and the target. Since classification accuracy is effectively lower-bounded by the chance level, panel B did not exhibit any negative impact of high feature similarity. However, when measuring performance by changes in cross-entropy loss, we observed results that align qualitatively with our theoretical predictions (panel C vs. Figs. 1D and 1F in the manuscript). Specifically, the combination of high feature similarity and low readout similarity resulted in negative knowledge transfer and retention. These effects were more pronounced when feature similarity was higher, given a fixed readout similarity, as predicted by our theory. Since the gradient was calculated using cross-entropy loss in the model, this observed effect is relevant to the continual learning of classification tasks, implying that our results are generalizable beyond regression conditions.

---

> > ### Comment · Reviewer_LdFf · 2024-08-11
> >
> > Thanks for your response! I’ll keep my score, and I believe this paper is above the acceptance bar for NeurIPS.

---

> > > ### Author Response · Authors · 2024-08-13
> > >
> > > Thank you again for your helpful comments.

---

### Official Review · Reviewer_Uj69 · 2024-07-11

**Soundness:** 3
**Presentation:** 3
**Contribution:** 3
**Rating:** 6
**Confidence:** 4

**Summary:**

The paper systematically investigated how feature similarity and readout similarity affect knowledge transfer and retention, under different gating scenarios, showing weight regularization based on Fisher information metric improves retention without compromising transfer. These are done with both linear teacher-student models and permuted MNIST task.

**Strengths:**

1) The paper addressed an important question in continual learning - task similarity - with very comprehensive experiments and successfully delineated the effect of several factors clearly; 2) the paper is clearly written; 3) although the bulk of the paper is done on linea teacher-student setting, the authors were able to solidify the majority of their results with permuted MNIST task.

**Weaknesses:**

I'm confused on the paper's definition of 'similarity' in general, for both feature similarity $\rho_a$ and readout similarity $\rho_b$. Since they are key concepts in the paper, they seem to deserve more careful treatment. See question.

**Questions:**

1) Could the authors please explain what they mean by 'element-wise correlation' for $\rho_a$ and $\rho_b$? When reading the task sampling procedure in appendix A.1, $\rho_a$ and $\rho_b$ are the probability of entries in the mixing matrices being identical; when deriving the analytical solutions, they seem to mean column-wise correlation? I could see either version serving as a working definition for feature similarity/readout similarity but just getting confused on having different definitions of key concepts in one paper.
2) Relatedly, could the authors comment on/discuss how different similarity definitions affect the conclusions in a broader setting? A third similarity definition seems to be used for the permuted MNIST task but similar trend could be observed. It seems worthwhile to explicitly address the stringiness of similarity definition in the discussion.

**Limitations:**

See question.

---

> ### Author Rebuttal · Authors · 2024-08-06
>
> Thank you for your insightful comments.
>
> __Weaknesses:__
> In panel A of the attached additional figure, we clarified the definitions of feature similarity and readout similarity in a classification setting. The green point in the left panel represents a scenario where two tasks have low feature similarity and high readout similarity: the input pixels are partially permuted while the output labels remain unchanged (Panel A, top). In contrast, the orange point indicates a scenario where tasks have high feature similarity and low readout similarity: the input pixels stay the same, but the output labels are partially permuted. The partial permutation of input pixels was achieved by randomly selecting a subset of input pixels with a probability of $1-\rho_a$, and then randomly permuting these selected pixels while keeping the remaining pixels fixed. The output permutation was performed in the same manner. A partial shift in target labels is expected to be common in real-world continual learning tasks in dynamic environments. Therefore, we believe it is crucial to understand how readout similarity influences continual learning performance alongside feature similarity.
>
> In panels B and C, we assessed the performance of knowledge transfer and retention for a continual classification task with partial input and label permutations. For this assessment, we employed the one-hidden layer model from Fig. 6 of the manuscript, with some modifications: a softmax function was added to the final layer, and cross-entropy loss was used for gradient calculation instead of mean-squared error. We observed that, even in this permuted continual classification task, our theory qualitatively explains the task similarity dependence of knowledge transfer and retention when they are measured by the cross-entropy loss (Panel C).
>
> __Question 1:__
> Thank you for pointing out this issue. The original manuscript indeed lacked clarity on this point. In our analytical estimation, we assumed that the elements of the input-projection matrices $A_1$ and $A_2$ were sampled jointly from a correlated Gaussian distribution. However, in our numerical verification, we generated $A_1$ and $A_2$ differently: we first sampled $A_1$ and then generated $A_2$ by replacing randomly selected elements of $A_1$ with independently sampled values from the same distribution, as depicted in Equation 10. Despite these differing sampling methods, they produce the same macroscopic behaviors due to the large $N_x$ assumption. For example, under Equation 10, the next-order term of Equation 32 changes slightly, resulting in the expression:
> $$\langle \lVert (D_1 A_1)^+ D_2 A_2 \rVert^2_F \rangle \approx
> N_s \left( \tilde{\alpha}^2 \rho_a^2 + \frac{\tilde{\alpha} N_s}{\alpha N_x} \left[ 1 + \frac{\rho_a}{N_s} (2 - \rho_a \alpha \tilde{\alpha}) \right] \right)$$
> Nevertheless, because the leading-order terms remain unchanged, our analytical results still align with the numerical results. We will clarify this point in the revised manuscript.
>
> __Question 2:__
> Our empirical results indicate that the derived analytical expression is robust against the choice of probability distribution for sampling matrices $A_1$ and $A_2$, provided they have zero mean and fixed covariance. However, higher-order moments appear in the analytical estimations of knowledge transfer and retention. Consequently, it remains uncertain whether the obtained result has a universality.
>
> Please note that the partial permutation introduced to MNIST tasks is methodologically similar to the resampling method depicted in Equation 10. Specifically, in the classification problem shown in the additional figure, we can set the latent space to be a 10-dimensional label space. In this scenario, the target projection matrix for the first task $B_1$ is a 10-dimensional identity matrix, while the matrix for the second task $B_2$ is generated by permuting a randomly selected submatrix of $B_1$.
>
> We will elaborate on these points in the discussion section of the revised manuscript.

---

> > ### Comment · Reviewer_Uj69 · 2024-08-12
> >
> > I would like to thank the authors for their clarification - all clear now. I would encourage the authors to add the rebuttal figure to their main text. I will increase my score to weak accept.

---

> > > ### Author Response · Authors · 2024-08-13
> > >
> > > Thank you for your positive evaluation. We will incorporate the rebuttal figure into the main text.

---

### Official Review · Reviewer_dowD · 2024-07-13

**Soundness:** 3
**Presentation:** 3
**Contribution:** 3
**Rating:** 6
**Confidence:** 3

**Summary:**

The paper investigates the challenge of continual learning in artificial neural networks, particularly when learning tasks with partial similarity. Task similarity can both facilitate knowledge transfer and increase the risk of interference and catastrophic forgetting. The authors develop a linear teacher-student model with latent structure to analyze how input feature similarity and readout pattern similarity affect knowledge transfer and forgetting. They find that high input feature similarity with low readout similarity is detrimental, while the opposite is relatively benign. The study also explores how different continual learning algorithms, such as task-dependent activity gating and weight regularization based on the Fisher information metric, interact with task similarity.

**Strengths:**

The paper stands out for its originality in several key areas: 1) Problem Formulation: The nuanced exploration of task similarity's impact on continual learning is a critical perspective. This focus on partial task similarity and its dual role in knowledge transfer and interference is novel and highly relevant. 2) Analytical Framework: The use of a linear teacher-student model with latent structure to dissect the effects of task similarity is an innovative approach. This model provides a clear, theoretical basis for understanding complex interactions in continual learning.
The quality of the research is evident in multiple dimensions: 1) Theoretical Rigor: The paper offers an analytical examination of task similarity's effects, supported by theoretical foundations. The linear teacher-student model is well-articulated and used to derive insightful conclusions. 2) Methodological Soundness: The evaluation of different continual learning algorithms, including task-dependent activity gating and weight regularization based on the Fisher information metric, is methodologically sound. These evaluations are carefully designed to highlight the algorithms' interactions with task similarity. 3) Empirical Validation: The empirical results on the permuted MNIST dataset provide strong support for the theoretical findings. The experiments are well-executed, with results that convincingly demonstrate the practical implications of the theoretical insights.

**Weaknesses:**

1. Limited Scope of Experimental Validation
While the paper provides theoretical insights and validates them on the permuted MNIST dataset, the experimental scope is somewhat limited. Validate the findings on a wider variety of datasets, including more complex and diverse datasets such as CIFAR-100, ImageNet, or non-visual tasks like language modeling or reinforcement learning environments. This would strengthen the generalizability of the results.
2. Insufficient Comparison with Existing Methods
The paper lacks a comprehensive comparison with a wider range of existing continual learning algorithms, particularly recent advancements in the field. Compare the proposed methods and insights with more recent state-of-the-art continual learning algorithms [1-6]. This includes methods that have been introduced in the past year, ensuring the evaluation is up-to-date.
[1]A comprehensive survey of continual learning: theory, method and application, 2024
[2]Lee S, Goldt S, Saxe A. Continual learning in the teacher-student setup: Impact of task similarity[C]//International Conference on Machine Learning. PMLR, 2021: 6109-6119.
[3]Lin S, Ju P, Liang Y, et al. Theory on forgetting and generalization of continual learning[C]//International Conference on Machine Learning. PMLR, 2023: 21078-21100.
[4]TRGP: Trust Region Gradient Projection for Continual Learning (ICLR ‘22)
[5]The Ideal Continual Learner: An Agent That Never Forgets (ICML ‘23)
[6]A Theoretical Study on Solving Continual Learning (NeurIPS ‘22)
3. Limited Discussion on Practical Implementation
The paper primarily focuses on theoretical insights and empirical validation but provides limited guidance on the practical implementation of the proposed methods. Include a section or supplementary material that provides detailed guidelines for implementing the proposed methods in practical scenarios. This could include code snippets, hyperparameter settings, and best practices.
4. Lack of In-Depth Analysis of Algorithm Interactions
While the paper evaluates task-dependent activity gating and weight regularization, it does not provide an in-depth analysis of how these methods interact with each other and with different levels of task similarity. Conduct experiments to analyze the interaction effects between different continual learning algorithms and varying degrees of task similarity. This would provide a deeper understanding of the conditions under which these methods are most effective.
5. Clarification on Theoretical Assumptions
The theoretical analysis is based on specific assumptions that may not always hold in practical scenarios. The paper could benefit from a more detailed discussion of these assumptions and their implications. Clearly state the assumptions underlying the theoretical model and discuss their limitations. Provide insights into how these assumptions might impact the applicability of the findings in real-world settings. Perform robustness analysis to test the sensitivity of the results to deviations from these assumptions. This could involve experimenting with variations in model architecture, data distribution, and task difficulty.

**Questions:**

Please refer to the Weaknesses above.

**Limitations:**

Please refer to the Weaknesses above.

---

> ### Author Rebuttal · Authors · 2024-08-06
>
> Thank you for your comments. Please find our replies to your comments on the weaknesses below.
>
> 1. In the attached figure, we demonstrated numerically that the feature and readout similarity influence the continual learning performance in a manner predicted by our theory even in a classification setting. We leave its application to more complex datasets and network architectures to future work.
>
> 2. We would like to seek clarification regarding this suggestion of comparing our work with “recent state-of-the-art continual learning algorithms [1-6]” and ensure our evaluation is up-to-date. Specifically, among the papers you cited, the first one is a review, while citations 2, 3, 5, and 6 are theoretical works that do not propose new algorithms. Could you please provide more details or specify the aspects of our work you believe need to be compared with these references?
>
> 3. Although proposing a new algorithm is not our primary focus, we did introduce an alternative approximation of weight regularization in the Fisher information metric that does not rely on diagonal approximation, unlike the elastic weight algorithm (Kirkpatrick et al., PNAS, 2017). We observed that this algorithm outperforms the elastic weight regularization method in a one-hidden-layer neural network when solving the MNIST task (Fig. 6E-H). While we presented the derivation of the alternative approximation specifically for a one-hidden-layer neural network (Eqs. 105-108 in Appendix E), this approach is applicable to fully-connected feedforward neural networks of any depth. We will provide a more general formulation in the revised manuscript to help practical implementation of the algorithm. Application of this approximation method to other neural architectures, such as convolutional, recurrent, or self-attention networks, is non-trivial and therefore will be addressed in future work.
>
> 4. While we did not explore the interaction between activity gating and weight regularization in this work, one interesting finding we obtained is their interchangeability in terms of knowledge transfer. Specifically, we demonstrated that Euclidean weight regularization with amplitude $\frac{N_x}{N_s} \left( \frac{1}{\gamma} - 1 \right)$ is equivalent to random activity gating with sparsity $\gamma$ (please see Sec. 6.1 for details). Here, $N_x$ and $N_s$ represent the widths of the input layer and latent source, respectively. This result suggests that combining gating and regularization is unlikely to provide additional benefits for knowledge transfer, but it may help prevent forgetting.
>
> 5. In Appendix A, we listed the assumptions introduced in our analysis. The first assumption, the random task assumption, posits that inputs and targets are generated randomly with a specific correlation structure. The second assumption, the low-dimensional latent assumption, imposes the existence of a low-dimensional latent space that generates both inputs and target outputs. Our derivation holds asymptotically, assuming that the input dimensionality is significantly larger than that of the latent space. We will provide further clarification in the main text of the revised manuscript.

---

> ### Comment · Reviewer_dowD · 2024-08-09
>
> Thank you for your detailed response. I appreciate the information provided and will increase the rating by 1 point. Regarding the references I mentioned, which focus on theoretical analysis of task similarity and continual performance, I hope to see some comparable analysis in the future.

---

> > ### Author Response · Authors · 2024-08-11
> >
> > Thank you for the clarification and positive recommendation. Please find the comparisons of our work with your references [2-6] below (we have omitted [1] as it is a survey article).
> >
> > Reference [2] (Lee et al., ICML 2021) is indeed closely related to our work, but we believe our study introduces two significant advancements. First, while their analysis of readout similarity and its comparison to input feature similarity was conducted numerically, we have derived analytical expressions for knowledge transfer and retention as functions of both feature and readout similarity. This approach uncovered non-trivial interactions between these two aspects of similarities. Secondly, their study focused on vanilla neural network training, whereas our work investigates the interaction between task similarity and widely used continual learning algorithms, such as activity gating and weight regularization, offering insights into their robustness against task similarity. Please note that we have provided a detailed comparison with reference [2] in the related work section (L88-L91) of the manuscript.
> >
> > The reference [3] (Lin et al., ICML 2023) is also related to our work. This work analyzed continual learning in a linear regression setting, derived the generalization error and the total forgetting in the presence of arbitrary number of tasks. In terms of the impact of task similarity, this work found that low task similarity may reduce forgetting, as observed in other theoretical works with different model settings (Lee et al., ICML 2021; Evron et al., CoLT 2022). In contrast, our work introduces a latent structure that allows us to disambiguate these two aspects. Additionally, their study did not explore the effects of continual learning algorithms such as activity gating and weight regularization.
> >
> > Reference [4] (Lin et al., ICLR 2022) presents a Trust-region gradient projection algorithm for continual learning. Unlike traditional approaches, this algorithm promotes forward knowledge transfer by projecting the gradient onto the weight space of previously learned, related tasks. Our work is motivated by a similar challenge—balancing the tradeoff between knowledge transfer and retention, a problem that remains not fully understood. In our study, we provide theoretical insights into how input and output similarity affect this tradeoff and how activity gating and weight regularization influence these interactions. We believe our findings will advance the development of algorithms in this area.
> >
> > References [5] (Peng et al., ICML 2023) and [6] (Kim et al., NeurIPS 2022) explore model-agnostic theories of continual learning. Peng et al. [5] investigated the necessary and sufficient conditions for achieving continual learning without forgetting. They demonstrated that, in a linear regression framework, an ideal continual learner could be constructed with respect to training error. However, their work does not address the effects of task similarity or specific learning algorithms. Similarly, Kim et al. [6] analyzed model-agnostic bounds in continual learning, with a focus on detecting task boundaries in a class incremental setting. Notably, like in many studies, our approach assumes that the task boundaries are known to the model, which makes their theoretical bounds less directly applicable to our scenario.
> >
> > We will discuss these works, particularly [3] and [4], in the related work section of the revised manuscript.

---

### Author Rebuttal · Authors · 2024-08-06

Thank you all for your valuable comments. Based on your feedback, we have added a figure that explains the feature and readout similarity in the context of a classification task (panel A) and shows their impact on knowledge transfer and retention (panels B and C). We found that when performance is measured by cross-entropy loss, our theoretical prediction explains the task similarity dependence of transfer and retention performance qualitatively even in a classification problem. Please see the individual replies for further details.

---

### Decision · Program_Chairs · 2024-09-25

**Decision:**

Accept (poster)

**Comment:**

This paper systematically investigates the impact of **feature similarity** and **readout similarity** on knowledge transfer and retention under different gating scenarios. It demonstrates that weight regularization based on the Fisher information metric can enhance retention without sacrificing transfer. The study is conducted using both linear teacher-student models and the permuted MNIST task.

The reviewers unanimously agree that the paper is well-motivated, with a novel theoretical formulation for the community. The analyses provided are thorough and valuable for understanding knowledge transfer dynamics. Given its contributions, the paper is considered above the acceptance threshold.

However, to further improve the paper's quality, the authors are requested to (1) provide a detailed clarification on the concepts of feature similarity and readout similarity to enhance understanding and (2) include additional experimental results in classification settings to validate the findings further.